# PRIME: Prioritizing Interpretability in Failure Mode Extraction

**Keivan Rezaei**[1*]**, Mehrdad Saberi**[1*]**, Mazda Moayeri**[1]**, Soheil Feizi**[1]

[1]Department of Computer Science, University of Maryland

{krezaei,msaberi,mmoayeri,sfeizi}@umd.edu

## Abstract

In this work, we study the challenge of providing human-understandable descriptions for failure modes in trained image classification models. Existing works address this problem by first identifying clusters (or directions) of incorrectly classified samples in a latent space and then aiming to provide human-understandable text descriptions for them. We observe that in some cases, describing text does not match well with identified failure modes, partially owing to the fact that shared interpretable attributes of failure modes may not be captured using clustering in the feature space. To improve on these shortcomings, we propose a novel approach that prioritizes interpretability in this problem: we start by obtaining human-understandable concepts (tags) of images in the dataset and then analyze the model's behavior based on the presence or absence of combinations of these tags. Our method also ensures that the tags describing a failure mode form a minimal set, avoiding redundant and noisy descriptions. Through several experiments on different datasets, we show that our method successfully identifies failure modes and generates high-quality text descriptions associated with them. These results highlight the importance of prioritizing interpretability in understanding model failures.

## 1 Introduction

A plethora of reasons (spurious correlations, imbalanced data, corrupted inputs, etc.) may lead a model to underperform on a specific subpopulation; we term this a *failure mode*. Failure modes are challenging to identify due to the black-box nature of deep models, and further, they are often obfuscated by common metrics like overall accuracy, leading to a false sense of security. However, these failures can have significant real-world consequences, such as perpetuating algorithmic bias (Buolamwini & Gebru, 2018) or unexpected catastrophic failure under distribution shift. Thus, the discovery and description of failure modes is crucial in building reliable AI, as we cannot fix a problem without first diagnosing it.

Detection of failure modes or biases within trained models has been studied in the literature. Prior work (Tsipras et al., 2020; Vasudevan et al., 2022) requires humans in the loop to get a sense of biases or subpopulations on which a model underperforms. Some other methods (Sohoni et al., 2020b; Nam et al., 2020; Kim et al., 2019; Liu et al., 2021) do the process of capturing and intervening in hard inputs without providing *human-understandable* descriptions for challenging subpopulations. Providing human-understandable and *interpretable* descriptions for failure modes not only enables humans to easily understand hard subpopulations, but enables the use of text-to-image methods (Ramesh et al., 2022; Rombach et al., 2022; Saharia et al., 2022; Kattakinda et al., 2022) to generate relevant images corresponding to failure modes to improve model's accuracy over them.

Recent work (Eyuboglu et al., 2022; Jain et al., 2022; Kim et al., 2023; d'Eon et al., 2021) takes an important step in improving failure mode diagnosis by additionally finding natural language descriptions of detected failure modes, namely via leveraging modern vision-language models. These methodologies leverage the shared vision-language latent space, discerning intricate clusters or directions within this space, and subsequently attributing human-comprehensible descriptions to them.

---

*Equal contribution.

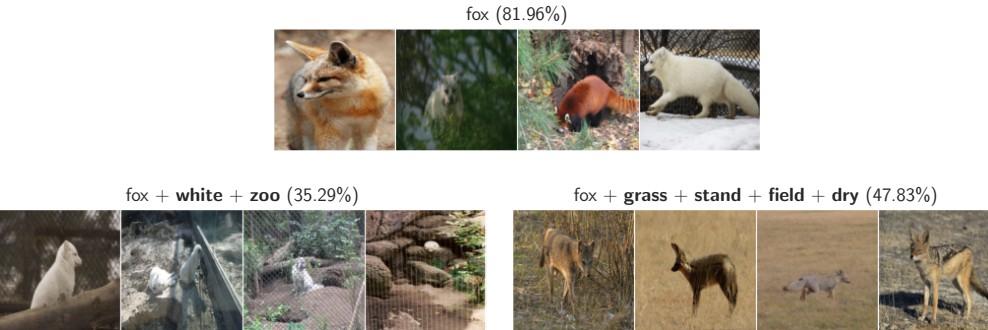

Figure 1: Visualization of two detected failure modes of class "fox" on a model trained on Living17. Overall accuracy for images of class "fox" is $81.96\%$. However, we identify two coherent subsets of images with significant accuracy drops: foxes standing in dry grass fields ($47.83\%$ accuracy) and foxes in a zoo where a white object (fox or other objects) is detected ($35.29\%$ accuracy). See Appendix A.4 for more examples.

However, questions have been raised regarding *the quality of the generated descriptions*, i.e., there is a need to ascertain whether the captions produced genuinely correspond to the images within the identified subpopulation. Additionally, it is essential to determine whether these images convey shared semantic attributes that can be effectively articulated through textual descriptions.

In this work, we investigate whether or not the latent representation space is a good proxy for semantic space. In fact, we consider two attributed datasets: CelebA(Liu et al., 2015) and CUB-200(Wah et al., 2011) and observe that two samples sharing many semantic attributes may indeed lie far away in latent space, while nearby instances may not share any semantics (see Section 5.2). Hence, existing methods may suffer from relying on representation space as clusters and directions found in this space may contain images with different semantic attributes leading to less coherent descriptions.

Inspired by this observation and the significance of faithful descriptions, we propose PRIME. In this method, we suggest to reverse the prevailing paradigm in failure mode diagnosis. That is, we put *interpretability first*. In our method, we start by obtaining human-understandable concepts (tags) of images using a pre-trained tagging model and examine model's behavior conditioning on the presence or absence of a combination of those tags. In particular, we consider different groups of tags and check whether (1) there is a significant drop in model's accuracy over images that represent all tags in the group and (2) that group is minimal, i.e., images having only some of those tags are easier images for the model. When a group of tags satisfies both of these conditions, we identify it as a failure mode which can be effectively described by these tags. Figure 2 shows the overview of our approach and compares it with existing methods.

As an example, by running PRIME on a trained model over Living17, we realize that images where a **black** ape is **hanging** from a **tree branch** identify a hard subpopulation such that model's accuracy drops from $86.23\%$ to $41.88\%$. Crucially, presence of all 3 of these tags is necessary, i.e., when we consider images that have 1 or 2 of these 3 tags, the accuracy of model is higher. Figure 3 illustrates these failure modes. We further study the effect of number of tags in Section 5.1.

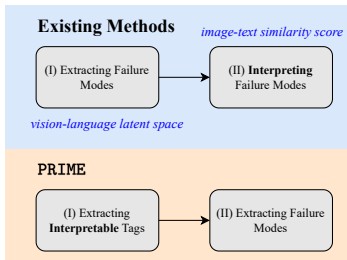

Figure 2: PRIME illustration.

To further validate our method, we examine data unseen during the computation of our failure mode descriptions. We observe that the images that match a failure mode lead the model to similarly struggle. That is, we demonstrate *generalizability* of our failure modes, crucially, directly from the succinct text descriptions. While reflecting the quality of our descriptions, this allows for bringing in generative models. We validate this claim by generating hard images using some of the failure mode's descriptions and compare the accuracy of model on them with some other generated images that correspond to easier subpopulations.

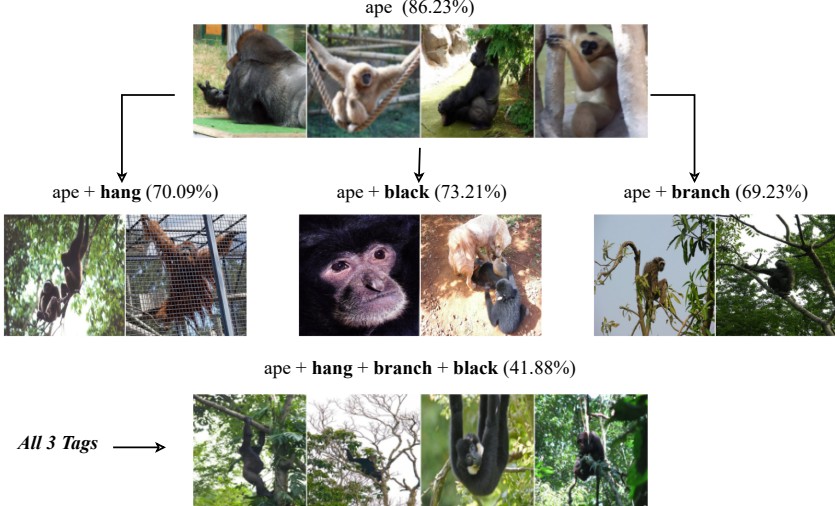

Figure 3: Although appearance of tags "hang", "black", and "branch" individually lowers model's accuracy, when all of them appear in the images, model's accuracy drops from 86.23% to 41.88%.

Finally, we show that PRIME produces better descriptions for detected failure modes in terms of *similarity*, *coherency*, and *specificity* of descriptions, compared to prior work that does not prioritize interpretability. Evaluating description quality is challenging and typically requires human assessment, which can be impractical for extensive studies. To mitigate that, inspired by CLIPScore (Hessel et al., 2021), we present a suite of three automated metrics that harness vision-language models to evaluate the quality. These metrics quantify both the intra-group image-description similarity and coherency, while also assessing the specificity of descriptions to ensure they are confined to the designated image groups. We mainly observe that due to putting interpretability first and considering different combinations of tags (concepts), we observe improvements in the quality of generated descriptions. We discuss PRIME's limitations in Appendix A.1.

**Summary of Contribution.**

1. We propose PRIME to extract and explain failure modes of a model in human-understandable terms by prioritizing interpretability.

2. Using a suite of three automated metrics to evaluate the quality of generated descriptions, we observe improvements in our method compared to strong baselines such as Eyuboglu et al. (2022) and Jain et al. (2022) on various datasets.

3. We advocate for the concept of putting interpretability first by providing empirical evidence derived from latent space analysis, suggesting that distance in latent space may at times be a misleading measure of semantic similarity for explaining model failure modes.

## 2 LITERATURE REVIEW

**Failure mode discovery.** The exploration of biases or challenging subpopulations within datasets, where a model's performance significantly declines, has been the subject of research in the field. Some recent methods for detecting such biases rely on human intervention, which can be time-consuming and impractical for routine usage. For instance, recent works (Tsipras et al., 2020; Vasudevan et al., 2022) depend on manual data exploration to identify failure modes in widely used datasets like ImageNet. Another line of work uses crowdsourcing (Nushi et al., 2018; Idrissi et al., 2022; Plumb et al., 2021) or simulators (Leclerc et al., 2022) to label visual features, but these methods are expensive and not universally applicable. Some researchers utilize feature visualization (Engstrom et al., 2019; Olah et al., 2017) or saliency maps (Selvaraju et al., 2017; Adebayo et al., 2018) to gain insights into the model's failure, but these techniques provide information specific to

individual samples and lack aggregated knowledge across the entire dataset. Some other approaches (Sohoni et al., 2020a; Nam et al., 2020; Liu et al., 2021; Hashimoto et al., 2018) automatically identify failure modes of a model but do not provide human-understandable descriptions for them.

Recent efforts have been made to identify difficult subpopulations and assign human-understandable descriptions to them (Eyuboglu et al., 2022; Jain et al., 2022; Kim et al., 2023). DOMINO (Eyuboglu et al., 2022) uses the latent representation of images in a vision-language model to cluster difficult images and then assigns human-understandable descriptions to these clusters. Jain et al. (2022) identifies a failure direction in the latent space and assigns description to images aligned with that direction. Hoffmann et al. (2021) shows that there is a semantic gap between similarity in latent space and similarity in input space, which can corrupt the output of methods that rely on assigning descriptions to latent embeddings. In (Kim et al., 2023), concepts are identified whose presence in images leads to a substantial decrease in the model's accuracy. Recent studies (Johnson et al., 2023; Gao et al., 2023) highlight the challenge of producing high-quality descriptions in the context of failure mode detection.

**Vision-Language and Tagging models.** Vision-language models have achieved remarkable success through pre-training on large-scale image-text pairs (Radford et al., 2021). These models can be utilized to incorporate vision-language space and evaluate captions generated to describe images. Recently Moayeri et al. (2023); Li et al. (2023) bridge the modality gap and enable off-the-shelf vision encoders to access shared vision-language space. Furthermore, in our method, we utilize models capable of generating tags for input images (Huang et al., 2023; Zhang et al., 2023).

## 3 EXTRACTING FAILURE MODES BY CONDITIONING ON HUMAN-UNDERSTANDABLE TAGS

Undesirable patterns or spurious correlations within the training dataset can lead to performance discrepancies in the learned models. For instance, in the Waterbirds dataset (Sagawa et al., 2019), images of landbirds are predominantly captured in terrestrial environments such as forests or grasslands. Consequently, a model can heavily rely on the background and make a prediction based on that. Conversely, the model may also rely on cues such as the presence of the ocean, sea, or boats to identify the input as waterbirds. This can result in performance drops for images where a waterbird is photographed on land or a landbird is photographed at sea. Detecting failure modes involves identifying groups of inputs where the model's performance significantly declines. While locating failure inputs is straightforward, *categorizing* them into distinct groups characterized by *human-understandable concepts* is a challenging task. To explain failure modes, we propose PRIME. Our method consists of two steps: (I) obtaining relevant tags for the images, and (II) identifying failure modes based on extracted tags.

### 3.1 OBTAINING RELEVANT TAGS

We start our method by collecting concepts (tags) over the images in the dataset. For example, for a photo of a fox sampled from ImageNet (Deng et al., 2009), we may collect tags "orange", "grass", "trees", "walking", "zoo", and others. To generate these tags for each image in our dataset, we employ the state-of-the-art *Recognize Anything Model (RAM)* (Zhang et al., 2023; Huang et al., 2023), which is a model trained on image-caption pairs to generate tags for the input images. RAM makes a substantial step for large models in computer vision, demonstrating the zero-shot ability to recognize any common category with high accuracy.

Let $\mathcal{D}$ be the set of all images. We obtain tags over all images of $\mathcal{D}$. Then, we analyze the effect of tags on prediction in a class-wise manner. In fact, the effect of tags and patterns on the model's prediction depends on the main object in the images, e.g., presence of water in the background improves performance on images labeled as waterbird while degrading performance on landbird images. For each class $c$ in the dataset, we take the union of tags generated by the model over images of class $c$. Subsequently, we eliminate tags that occur less frequently than a predetermined threshold. This threshold varies depending on the dataset size, specifically set at 50, 100, and 200 in our experimental scenarios. In fact, we remove rare (irrelevant) tags and obtain a set of tags $T_c$ for each class $c$ in the dataset, e.g., $T_c = \{$ "red", "orange", "snow", "grass", ...$\}$.

## 3.2 DETECTING FAILURE MODES

After obtaining tags, we mainly focus on tags whose presence in the image leads to a performance drop in the model. Indeed, for each class $c$, we pick a subset $S_c \subseteq T_c$ of tags and evaluate the model's performance on the images of class $c$ including all tags in $S_c$. We denote this set of images by $I_{S_c}$. $I_{S_C}$ is a coherent image set in the sense that those images share at least the tags in $S_c$.

For $I_{S_c}$, to be a *failure mode*, we require that the model's accuracy over images of $I_{S_c}$ significantly drops, i.e., denoting the model's accuracy over images of $I_{S_c}$ by $A_{S_c}$ and the model's overall accuracy over the images of class $c$ by $A_c$, then $A_{S_c} \leq A_c - a$. Parameter $a$ plays a pivotal role in determining the severity of the failure modes we aim to detect. Importantly, we want the tags in $S_c$ to be minimal, i.e., none of them should be redundant. In order to ensure that, we expect that the removal of any of tags in $S_c$ determines a relatively easier subpopulation. In essence, presence of all tags in $S_c$ is deemed essential to obtain that hard subpopulation.

More precisely, Let $n$ to be the cardinality of $S_c$, i.e., $n = |S_c|$. We require all tags $t \in S_c$ to be necessary. i.e., if we remove a tag $t$ from $S_c$, then the resulting group of images should become an easier subpopulation. More formally, for all $t \in S_c$, $A_{S_c \setminus t} \geq A_{S_c} + b_n$ where $b_2, b_3, b_4, ...$ are some hyperparameters that determine the degree of necessity of appearance of all tags in a group. We generally pick $b_2 = 10\%$, $b_3 = 5\%$ and $b_4 = 2.5\%$ in our experiments. These values help us fine-tune the sensitivity to tag necessity and identify meaningful failure modes. Furthermore, we require a minimum of $s$ samples in $I_{S_c}$ for reliability and generalization. This ensures a sufficient number of instances where the model's performance drops, allowing us to confidently identify failure modes. Figure 1 shows some of the obtained failure modes.

**How to obtain failure modes.** We generally use *Exhaustive Search* to obtain failure modes. In exhaustive search, we systematically evaluate various combinations of tags to identify failure modes, employing a brute-force approach that covers all possible combinations of tags up to $l$ ones. More precisely, we consider all subsets $S_c \subseteq T_c$ such that $|S_c| \leq l$ and evaluate the model's performance on $I_{S_c}$. As mentioned above, we detect $S_c$ as a failure mode if (1) $|I_{S_c}| \geq s$, (2) model's accuracy over $I_{S_c}$ is at most $A_c - a$, and (3) $S_c$ is minimal, i.e., for all $t \in S_c$, $A_{S_c \setminus t} \geq A_{S_c} + b_{|S_c|}$. It is worth noting that the final output of the method is all sets $I_{S_c}$ that satisfy those conditions and **description** for this group consist of **class name** ($c$) and **all tags in** $S_c$.

We note that the aforementioned method runs with a complexity of $O\left(|T_c|^l |\mathcal{D}|\right)$. However, $l$ is generally small, i.e., for a failure mode to be generalizable, we mainly consider cases where $l \leq 4$. Furthermore, in our experiments over different datasets $|T_c| \approx 100$, thus, the exhaustive search is relatively efficient. For instance, running exhaustive search ($l = 4, s = 30, a = 30\%$) on Living17 dataset having 17 classes with 88400 images results in obtaining 132 failure modes within a time frame of under 5 minutes. We refer to Appendix A.6 for more efficient algorithms and Appendix A.15 for more detailed explanation of `PRIME`'s hyperparameters.

**Experiments and Comparison to Existing Work.** We run experiments on models trained on Living17, NonLiving26, Entity13 (Santurkar et al., 2020), Waterbirds (Sagawa et al., 2019), and CelebA (Liu et al., 2015) (for age classification). We refer to Appendix A.2 for model training details and the different hyperparameters we used for failure mode detection. We refer to Appendix A.3 for the full results of our method on different datasets. We engage two of the most recent failure mode detection approaches DOMINO(Eyuboglu et al., 2022) and Distilling Failure Directions(Jain et al., 2022) as strong baselines and compare our approach with them.

## 4 EVALUATION

Let $\mathcal{D}$ be the dataset on which we detect failure modes of a trained model. The result of a human-understandable failure mode extractor on this dataset consists of sets of images, denoted as $I_1, I_2, ..., I_m$, along with corresponding descriptions, labeled as $T_1, T_2, ..., T_m$. Each set $I_j$ comprises images that share similar attributes, leading to a noticeable drop in model accuracy. Number of detected failure modes, $m$, is influenced by various hyperparameters, e.g., in our method, minimum accuracy drop ($a$), values for $b_2, b_3, ...$, and the minimum group size ($s$) are these parameters.

One of the main goals of detecting failure modes in human-understandable terms is to generate high-quality captions for hard subpopulations. We note that these methods should also be evaluated

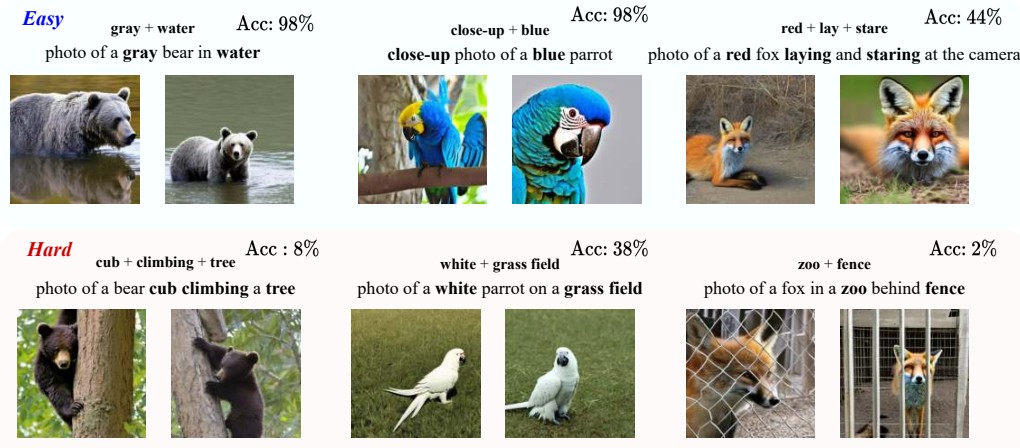

Figure 4: Accuracy of model over 50 generated images corresponding to one of the success modes and failure modes for classes "bear", " parrot", and "fox" from Living17. Accuracy gap shows that our method can identify hard and easy subpopulations. Images show that extracted tags are capable of describing detailed images.

in terms of coverage, i.e., what portion of failure inputs are covered along with the performance gap in detected failure modes. All these methods extract hard subpopulations on which the model's accuracy significantly drops, and coverage depends on the dataset and the hyperparameters of the method, thus, we mainly focus on generalizability of our approach and quality of descriptions.

## 4.1 GENERALIZATION ON UNSEEN DATA

In order to evaluate generalizability of the resulted descriptions, we take dataset $\mathcal{D}'$ including unseen images and recover relevant images in that to each of captions $T_1, T_2, ..., T_m$, thus, obtaining $I'_1, I'_2, ..., I'_m$. Indeed, $I'_j$ includes images in $\mathcal{D}'$ that are relevant to $T_j$. If captions can describe hard subpopulations, then we expect hard subpopulations in $I'_1, I'_2, ..., I'_m$. Additionally, since $\mathcal{D}$ and $\mathcal{D}'$ share the same distribution, we anticipate the accuracy drop in $I_j$ to closely resemble that in $I'_j$.

In our method, for a detected failure mode $I_{S_c}$, we obtain $I'_{S_c}$ by collecting images of $\mathcal{D}'$ that have all tags in $S_c$. For example, if appearance of tags "black", "snowing", and "forest" is detected as a failure mode for class "bear", we evaluate model's performance on images of "bear" in $\mathcal{D}'$ that include those three tags, expecting a significant accuracy drop for model on those images. As seen in Figure 9, PRIME shows a good level of generalizability. We refer to Appendix A.5 for generalization on other datasets with respect to different hyperparameters ($s$ and $a$). While all our detected failure modes generalize well, we observe stronger generalization when using more stringent hyperparameter values (high $s$ and $a$), though it comes at the cost of detecting fewer modes.

In contrast, existing methods (Eyuboglu et al., 2022; Jain et al., 2022) do not provide a direct way to assess generalization from text descriptions alone. See Appendix A.9 for more details.

## 4.2 GENERALIZATION ON GENERATED DATA

In this section we validate PRIME on synthetic images. To utilize image generation models, we employ language models to create descriptive captions for objects and tags associated with failure modes in images. We note that use of language models is just for validation on synthetic images, it is not a part of PRIME framework. We discuss about limitations of using language models in Appendix A.1. These captions serve as prompts for text-to-image generative models, enabling the creation of artificial images that correspond to the identified failure modes. To achieve this, we adopt the methodology outlined in Vendrow et al. (2023), which leverages a denoising diffusion model (Ho et al., 2020; Rombach et al., 2022). We fine-tune the generative model on the Living17 dataset to generate images that match the distribution of the data that the classifiers is trained on.

For each class in Living17 dataset, we apply our approach to identify two failure modes (hard subpopulations) and two success modes (easy subpopulations). We then employ ChatGPT[1] to generate descriptive captions for these groups. Subsequently, we generate 50 images for each caption and assess the model's accuracy on these newly generated images. We refer to Appendix A.10 for more details on this experiment and average discrepancy in accuracy between the success modes and failure modes which further validates PRIME. Figure 4 provides both accuracy metrics and sample images for three hard and three easy subpopulations.

## 4.3 Quality of Descriptions

Within this section, our aim is to evaluate the quality of the descriptions for the identified failure modes. In contrast to Section 4.2, where language models were employed to create sentence descriptions using the tags associated with each failure mode, here we combine tags and class labels in a bag-of-words manner. For instance, when constructing the description for a failure mode in the "ape" class with the tags "black" + "branch," we formulate it as "a photo of ape black branch". We discuss more about it in Appendix A.1.

In order to evaluate the quality of descriptions, we propose a suite of three complementary automated metrics that utilize vision-language models (such as CLIP) as a proxy to obtain image-text similarity (Hessel et al., 2021). Let $t$ be the failure mode's description, $f_{\text{text}}(t)$ denote the normalized embedding of text prompt $t$ and $f_{\text{vision}}(x)$ denote the normalized embedding of an image $x$. The similarity of image $x$ to this failure mode's description $t$ is the dot product of image and text representation in shared vision-language space. More precisely, $\text{sim}(x, t) := \langle f_{\text{vision}}(x), f_{\text{text}}(t) \rangle$.

For a high-quality failure mode $I_j$ and its description $T_j$, we wish $T_j$ to be similar to images in $I_j$, thus, we consider the average *similarity* of images in $I_j$ and $T_j$. we further expect a high level of *coherency* among all images in $I_j$, i.e., these images should all share multiple semantic attributes described by text, thus, we wish the standard deviation of similarity scores between images in $I_j$ and $T_j$ to be low. Lastly, we expect generated captions to be *specific*, capturing the essence of the failure mode, without including distracting irrelevant information. That is, caption $T_j$ should only describe images in $I_j$ and not images outside of that. As a result, we consider the AUROC between the similarity score of images inside the failure mode ($I_j$) and some randomly sampled images outside of that. We note that in existing methods as well as our method, all images in a failure mode have the same label, so we sample from images outside of the group but with the same label.

In Figure 5, we show (1) the average similarity score, i.e., for all $I_j$ and $x \in I_j$, we take the mean of $\text{sim}(x, T_j)$, (2) the standard deviation of similarity score, i.e., the standard deviation of $\text{sim}(x, T_j)$ for all $I_j$ and $x \in I_j$, and (3) the AUROC between the similarity scores of images inside failure modes to their corresponding description and some randomly sampled images outside of the failure mode to that. As shown in Figure 5, PRIME improves over DOMINO (Eyuboglu et al., 2022) in terms of all AUROC, average similarity, and standard deviation on different datasets. It is worth noting that this improvement comes even though DOMINO chooses a text caption for the failure mode ***to maximize the similarity score in latent space***. We use hyperparameters for DOMINO to obtain fairly the same number of failure modes detected by PRIME. Results in Figure 5 show that PRIME is better than DOMINO in the descriptions it provides for detected failure modes. In Appendix A.8 we provide more details on these experiments. Due to the limitations of Jain et al. (2022) for automatically generating captions, we cannot conduct extensive experiments on various datasets. More details and results on that can be found in Appendix A.11.

## 5 On Complexity of Failure Mode Explanations

We note that the main advantage of our method is its more faithful interpretation of failure modes. This comes due to (1) putting interpretability first, i.e., we start by assigning interpretable tags to images and then recognize hard subpopulations and (2) considering combination of several tags which leads to a higher number of attributes (tags) in the description of the group.

---

[1]ChatGPT 3.5, August 3 version

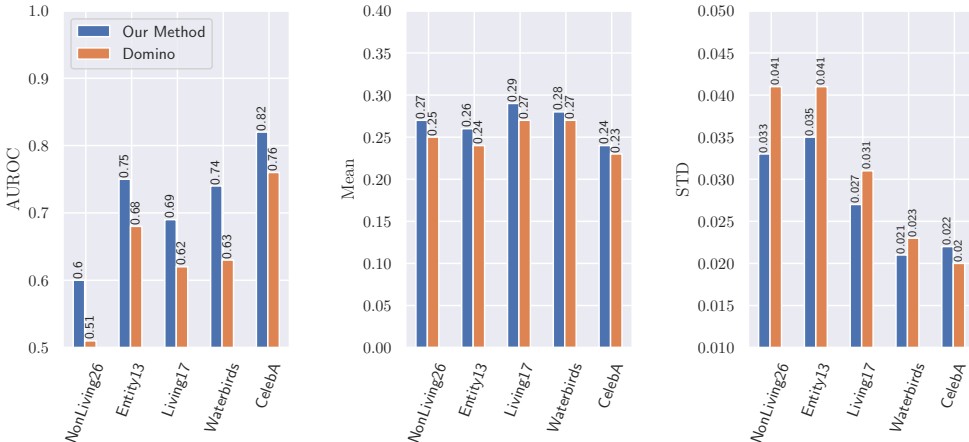

Figure 5: The mean and standard deviation of similarity scores between images in failure modes and their respective descriptions, along with the AUROC measuring the similarity score between descriptions and images inside and outside of failure modes, demonstrate that our method outperforms DOMINO in descriptions it generates for detected failure modes across various datasets.

## 5.1 DO WE NEED TO CONSIDER COMBINATION OF TAGS?

We shed light on the number of tags in the failure modes detected by our approach. We note that unlike Bias2Text (Kim et al., 2023) that finds biased concepts on which model's behavior changes, we observe that sometimes appearance of several tags (concepts) all together leads to a severe failure mode. As an example, we refer to Table 1 where we observe that appearance of all 3 tags together leads to a significant drop while single tags and pairs of them show relatively better performance.

Table 1: Accuracy on unseen images ($\mathcal{D}'$) for class "ape" when given tags appear in the inputs (see Table 5 for visualization).

| # of Tags | Tags | Accuracy |
|---|---|---|
| 3 | hang; branch; black; | 41.18% |
| 2 | hang; branch; | 56.33% |
| | hang; black; | 56.25% |
| | branch; black; | 54.67% |
| 1 | hang; | 70.09% |
| | branch; | 69.23% |
| | black; | 73.21% |

Table 2: Average accuracy drop over unseen images ($\mathcal{D}'$) on failure modes with 3 tags and images have at least 2 of those tags or at least one of them.

| Dataset | 3 Tags | 2 Tags | 1 Tag |
|---|---|---|---|
| Entity13 | 34.75% | 25.29% | 14.86% |
| Living17 | 26.82% | 17.13% | 8.18% |
| Waterbirds | 23.35% | 14.43% | 7.19% |
| CelebA | 23.25% | 16.84% | 9.02% |

In `PRIME`, we emphasize the necessity of tags. Specifically, for any detected failure mode, the removal of any tag would result in an easier subpopulation. Consequently, failure modes with more tags not only provide more detailed description of their images but also characterize more challenging subpopulations. Table 2 presents the average accuracy drop on unseen images for groups identified by three tags, compared to the average accuracy drop on groups identified by subsets of two tags or even a single tag from those failure modes. These results clearly demonstrate that involving more tags leads to the detection of more challenging subpopulations.

## 5.2 CLUSTERING-BASED METHODS MAY STRUGGLE IN GENERATING COHERENT OUTPUT

We empirically analyze the reverse direction of detecting human-understandable failure modes. We note that in recent work where the goal is to obtain interpretable failure modes, those groups are found by clustering images in the latent space. Then, when a group of images or a direction in the latent space is found, these methods leverage the shared space of vision-language to find the text that best describes the images inside the group.

We argue that these approaches, based on distance-based clusters in the representation space, may produce less detailed descriptions. This is because the representation space doesn't always align perfectly with the semantic space.

Table 3: Statistics of the distance between two points in CelebA conditioned on number of shared tags. Distances are reported using CLIP ViT-B/16 representation space. The last column shows the probability that the distance between two sampled images with at least $d$ common tags be more than that of two randomly sampled images.

| # of shared tags $\geq d$ | mean | standard deviation | Probability |
|---|---|---|---|
| $d = 0$ | 9.49 | 0.98 | 0.50 |
| $d = 1$ | 9.47 | 1.00 | 0.49 |
| $d = 3$ | 9.23 | 1.00 | 0.42 |
| $d = 5$ | 8.89 | 1.21 | 0.34 |
| $d = 7$ | 8.32 | 1.80 | 0.25 |

Even points close to each other in the feature space may differ in certain attributes, and conversely, points sharing human-understandable attributes may not be proximate in the feature space. Hence, these approaches cannot generate high-quality descriptions as their detected clusters in the representation space may contain images with other semantic attributes.

To empirically test this idea, we use two attribute-rich datasets: CelebA (Liu et al., 2015) and CUB-200 (Wah et al., 2011). CelebA features 40 human-understandable tags per image, while CUB-200, a dataset of birds, includes 312 tags per image, all referring to semantic attributes. We use CLIP ViT-B/16 (Radford et al., 2021) and examine its representation space in terms of datasets' tags. Table 3 shows the statistics of the distance between the points conditioned on the number of shared tags. As seen in the Table 3, although the average of distance between points with more common tags slightly decreases, the standard deviation of distance between points is high. In fact, points with many common tags can still be far away from each other. Last column in Table 3 shows the probability that the distance between two points with at least $d$ shared tags be larger than the distance of two randomly sampled points. Even when at least 5 tags are shared between two points, with the probability of 0.34, the distance can be larger than two random points. Thus, if we plant a failure mode on a group of images sharing a subset of tags, these clustering-based methods cannot find a group consisting of *only* those images; they will inevitably include other irrelevant images, leading to an incoherent failure mode set and, consequently, a low-quality description. This can be observed in Appendix A.7 where we include DOMINO's output.

We also run another experiment to foster our hypothesis that distance-based clustering methods cannot fully capture semantic similarities. We randomly pick an image $x$ and find $N$ closest images to $x$ in the feature space. Let $C$ be the set of these images. We inspect this set in terms of the number of tags that commonly appear in its images as recent methods (Eyuboglu et al., 2022; d'Eon et al., 2021; Jain et al., 2022), take the average embedding of images in $C$ and then assign a text to describe images of $C$. Table 6 shows the average number of tags that appear in at least $\alpha N$ images of set $C$ (we sample many different points $x$). If representation space is a good proxy for semantic space, then we expect a large number of shared tags in close proximity to point $x$. At the same time, for the point $x$, we find the maximum number of tags that appear in $x$ and at least $N$ other images. This is the number of shared tags in close proximity of point $x$ but in semantic space. As shown in Table 6, average number of shared tags in semantic space is significantly larger than the average number of shared tags in representation space.

## 6 CONCLUSIONS

In this study, drawing from the observation that current techniques in human-comprehensible failure mode detection sometimes produce incoherent descriptions, along with empirical findings related to the latent space of vision-language models, we introduced PRIME, a novel approach that prioritizes interpretability in failure mode detection. Our results demonstrate that it generates descriptions that are more similar, coherent, and specific compared to existing methods for the detected failure modes.

ACKNOWLEDGMENTS

This project was supported in part by a grant from an NSF CAREER AWARD 1942230, ONR YIP award N00014-22-1-2271, ARO's Early Career Program Award 310902-00001, Meta grant 23010098, HR00112090132 (DARPA/RED), HR001119S0026 (DARPA/GARD), Army Grant No. W911NF2120076, NIST 60NANB20D134, the NSF award CCF2212458, an Amazon Research Award and an award from Capital One.

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

# A  APPENDIX

## A.1  LIMITATIONS

**Utilizing pre-trained tagging models**  We note that in this work, we used RAM (Zhang et al., 2023) to extract tags from images and then used those tags to identify hard subpopulations. Accuracy of RAM which is the state-of-the-art tagging model is well studied in (Zhang et al., 2023). Authors observe high-quality performance of their method on different common datasets over different tasks. We also conducted a small-scale human validation study to measure the accuracy of RAM on Living17 dataset (see Appendix A.14 for more details) and observe strong performance. However, we acknowledge that PRIME, like other existing methods, utilizes an auxiliary model to bridge the gap between vision and language modalities. Thus, its performance is affected and limited by the auxiliary method. Notably, our use of the auxiliary model (tagging models) is closely aligned to the task they are optimized for. That is, these models are trained to efficiently detect various objects and concepts in images, assigning tags accordingly. As tagging models advance, PRIME's effectiveness is expected to be enhanced.

**PRIME for specific domains**  We note that PRIME relies on tagging model to extract informative tags from image, making it less effective on specific domains for which the tagging model is not optimally suited. However, we believe it is highly likely that tagging models like RAM that are finetuned on specific domains will arise very soon, similar to how ConVIRT (CLIP for medical data) (Zhang et al., 2022) came about soon after CLIP. Leveraging finetuned tagging models, PRIME remains effective for failure mode extraction on specific domains such as medical images.

**Tag interpretation**  Acknowledging potential challenges in interpreting certain tags within PRIME, particularly adjectives, is imperative. For instance, the tag "white" may not consistently align with the primary object in the image, as its detection could be influenced by background elements or other objects present. Despite this, the utility of PRIME persists, serving as a valuable tool to highlight instances where model failures occur at the convergence of specific tags, even if not exclusively tied to the main object in the image. We note that, due to this limitation, we do not use language models to generate descriptions from tags (except for validating PRIME with image generation) because language models may associate tags with class labels, which is not necessarily correct. Therefore, in evaluating the quality of descriptions, we used a bag-of-word manner to obtain PRIME's results.

## A.2  TRAINING MODELS

We train a model on each dataset we are considering.

- For ImageNet, we use the standard pretrained ResNet50 model (He et al., 2015).
- For Living17, Entity13, and NonLiving26, we utilize a DINO self-supervised model (Caron et al., 2021) with ResNet50 backbone and fine-tune it over Living17. we used SGD with following hyperparameters to finetune the model for 5 epochs.
    - lr = 0.001
    - momentum = 0.9
- For CelebA (age classification), we used a pretrained ResNet18 model (He et al., 2015) which is finetuned for 5 epochs using SGD the following hyperparameters.
    - lr = 0.001
    - momentum = 0.9
    - weight decay = $5e - 4$

  We note that classifier is trained in a way that it is biased toward images of young women and old men.
- For Waterbirds, we fine-tuned a pretrained ResNet18 model (He et al., 2015) for 20 epoch using SGD the following hyperparameters.
    - lr = 0.001

> – momentum $= 0.9$
> – weight decay $= 5e - 4$

## A.3 DETAILED RESULTS OF DETECTED FAILURE MODES

Our method on **Living17**:
**Results**: 36 failure modes with 1 tag, 68 failure modes with 2 tags, 24 failure modes with 3 tags, and 4 failure modes with 4 tags.
**Hyperparameters**: $s = 30$, $a = 30$, $b_2 = 10\%$, $b_3 = 5\%$, and $b_4 = 2.5\%$.

- class "wolf" (Accuracy: 83.69%):
  hide (54.86%); —- floor + hide (38.71%); —- floor + hide (38.71%); —- den (49.06%); —- den + hide (22.22%); —- lay + red (41.86%); —- den + lay (29.27%); —- hide + lay (41.51%); —- night (44.44%); —- floor + cub (48.48%); —- floor + den + cub (22.22%); —- grass + stare + hide + red (66.67%); —- log + red (62.86%); —- grass + tree + brown (63.64%);
- class "cat" (Accuracy: 89.58%): enclosure (52.63%); —- zoo (50.00%); —- habitat (34.09%); —- grassy (63.64%); —- tiger + walk (28.57%); —- tiger + grass (58.54%); —- bengal tiger + walk (28.57%); —- bengal tiger + grass (60.00%); —- floor + tree (57.14%); —- log (56.76%); —- white + grass (63.64%); —- tiger + tree (37.50%); —- hide + stand (82.50%);

Our method on **Entity13**:
**Results**: 45 failure modes with 1 tag, 45 failure modes with 2 tags, and 18 failure modes with 3 tags.
**Hyperparameters**: $s = 100$, $a = 30$, $b_2 = 10\%$, $b_3 = 5\%$, and $b_4 = 2.5\%$.

- class "wheeled vehicle" (Accuracy: 88.05%):
  shopping cart (53.03%); —- floor + cart (54.17%); —- sit + shopping cart (43.90%); —- cage (44.20%); —- basket (50.45%); —- man + pole (65.79%);
- class "produce, green goods, green groceries, garden truck" (Accuracy: 92.91%): floor + food (74.77%);
- class "accessory, accoutrement, accouterment" (Accuracy: 63.98%): swimwear + pose (18.18%); —- stand + pose + black (31.15%); —- brunette (19.23%); —- swimwear + brunette (4.20%); —- person + graduation (33.92%);

Our method on CelebA (Young vs. Old classification):
**Results**: 45 failure modes with 1 tag, 27 failure modes with 2 tags, and 11 failure modes with 3 tags.
**Hyperparameters**: $s = 100$, $a = 30$, $b_2 = 10\%$, $b_3 = 5\%$, and $b_4 = 2.5\%$.

- class "young" (Accuracy: 80%):
  beard (32.34%); —- man + laugh (41.21%); —- smile + tie + stand (58.27%); —- man + goggles (41.04%); —- man + sunglasses (36.11%); —- man + white + stand (69.32%); —- man + sing (33.78%); —- man + microphone (40.00%); —- business suit + smile + stand (51.11%); —- black + goggles (42.65%);

Our method on **CelebA** (Young vs Old classification):
**Results**: 45 failure modes with 1 tag, 27 failure modes with 2 tags, and 11 failure modes with 3 tags.
**Hyperparameters**: $s = 100$, $a = 30$, $b_2 = 10\%$, $b_3 = 5\%$, and $b_4 = 2.5\%$.

- class "young" (Accuracy: 80%):
  beard (32.34%); —- man + laugh (41.21%); —- smile + tie + stand (58.27%); —- man + goggles (41.04%); —- man + sunglasses (36.11%); —- man + white + stand (69.32%); —- man + sing (33.78%); —- man + microphone (40.00%); —- business suit + smile + stand (51.11%); —- black + goggles (42.65%);

Our method on **Waterbirds**:
**Results**: 4 failure modes with 1 tag, 8 failure modes with 2 tags, and 9 failure modes with 3 tags.
**Hyperparameters**: $s = 100$, $a = 30$, $b_2 = 10\%$, $b_3 = 5\%$, and $b_4 = 2.5\%$.

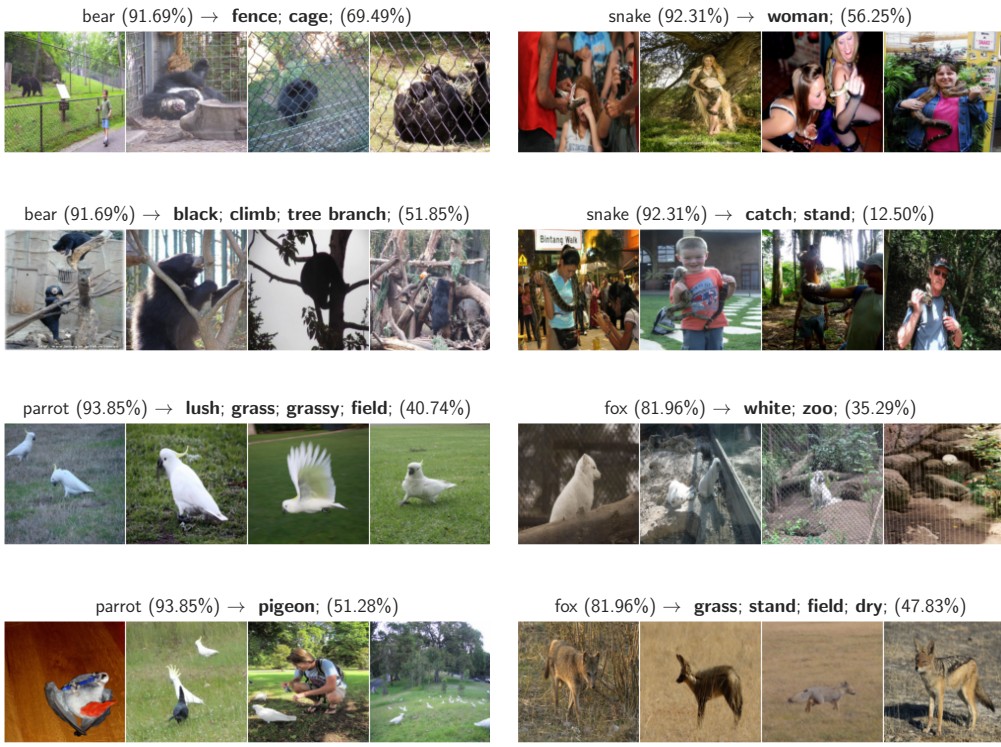

Figure 6: Some visualization of detected failure modes on Living17.

- class "landbird" (Accuracy: $87.41\%$):
  black + sea ($62.50\%$); —- crow + water ($64.58\%$); —- water + person ($69.86\%$); —- black + water + beak ($71.43\%$); —- water + man ($71.93\%$); —- stand + sea + blue ($65.71\%$); —- sea + sit + boat ($75.68\%$); —- black + ledge + sea ($58.82\%$);

- class "waterbird" (Accuracy: $33.00\%$): wood ($10.47\%$); —- stem ($6.67\%$); —- stand + tree + pole ($16.67\%$);

## A.4 VISUALIZATION ON SOME OF THE DETECTED FAILURE MODES

We refer to Figures 6, 7, and 8 for more visualization of failure modes detected in our approach on Living17, ImageNet, and Waterbirds.

## A.5 RUNNING THE METHOD USING DIFFERENT VALUES OF $(s, a)$

In this section, we inspect the effect of different hyperparameters $(s, a)$ on the result of our method. By increasing $a$, we aim to detect harder subpopulations, thus, the number of detected failure modes will decrease. By increasing $s$, we detect a group of images associated with a set of tags as a failure mode, if there are a significant number of images within that group. This brings more generalization over detected failure modes while a fewer number of them will be detected by the method. Figure 10 shows the generalization plot over different datasets with respect to different hyperparameters.

In Table 4, we also report correlation coefficien between train drop and test drop of failure modes over different datasets and values of $s$ and $a$.

## A.6 GREEDY SEARCH

We note that in our experiments, exhaustive search was efficient enough so that we do not need to consider any other approaches. We used some heuristic approaches to improve the efficiency of exhaustive search such as eliminating combination of tags that a few images represent them, etc.

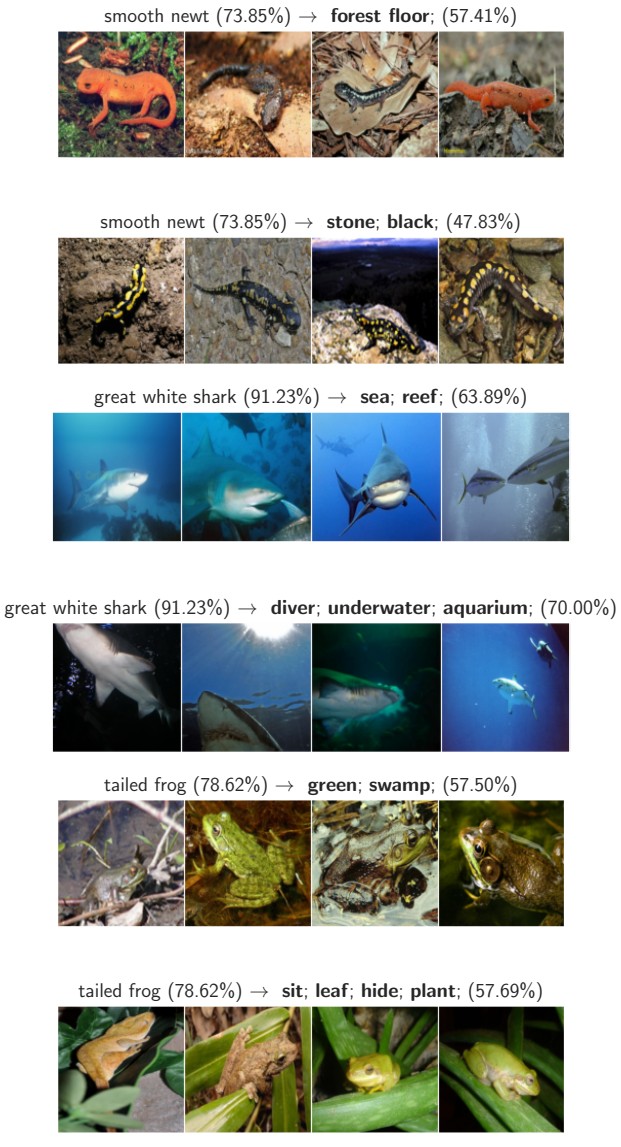

Figure 7: Some visualization of detected failure modes on ImageNet.

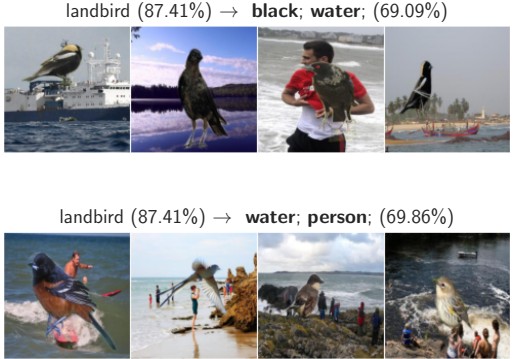

Figure 8: Some visualization of detected failure modes on Waterbirds.

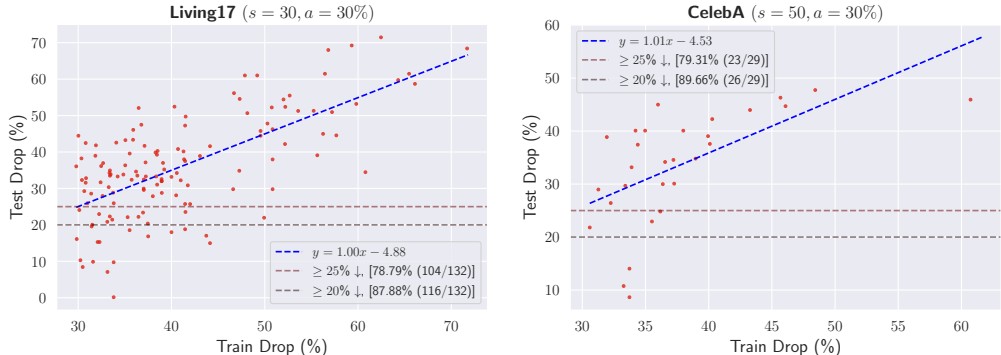

Figure 9: Evaluating detected failure modes on unseen data. (**Left**): we extract failure modes on Living17 dataset using $s = 30$ and $a = 30\%$. 132 failure groups (over 17 classes) are detected and it is observed that around $86.01\%$ of detected failure modes exhibit at least $25\%$ drop in accuracy over unseen data that shows a significant degree of generalization. (**Right**): same results for CelebA dataset where the parameters for failure mode detection is $s = 50$ and $a = 30\%$. Around $79.31\%$ of failure modes show the drop of at least $20\%$. The trend of $y = x$ is seen in these plots.

| Dataset | $s$ | $a$ | corrcoef |
|---|---|---|---|
| | 25 | 25 | 0.70 |
| Living17 | 50 | 30 | 0.73 |
| | 30 | 30 | 0.72 |
| | 100 | 30 | 0.73 |
| Entity13 | 200 | 30 | 0.73 |
| | 200 | 25 | 0.85 |
| | 50 | 30 | 0.60 |
| CelebA | 100 | 30 | 0.83 |
| | 100 | 25 | 0.81 |

Table 4: Correlation coefficien between train drop and test drop.

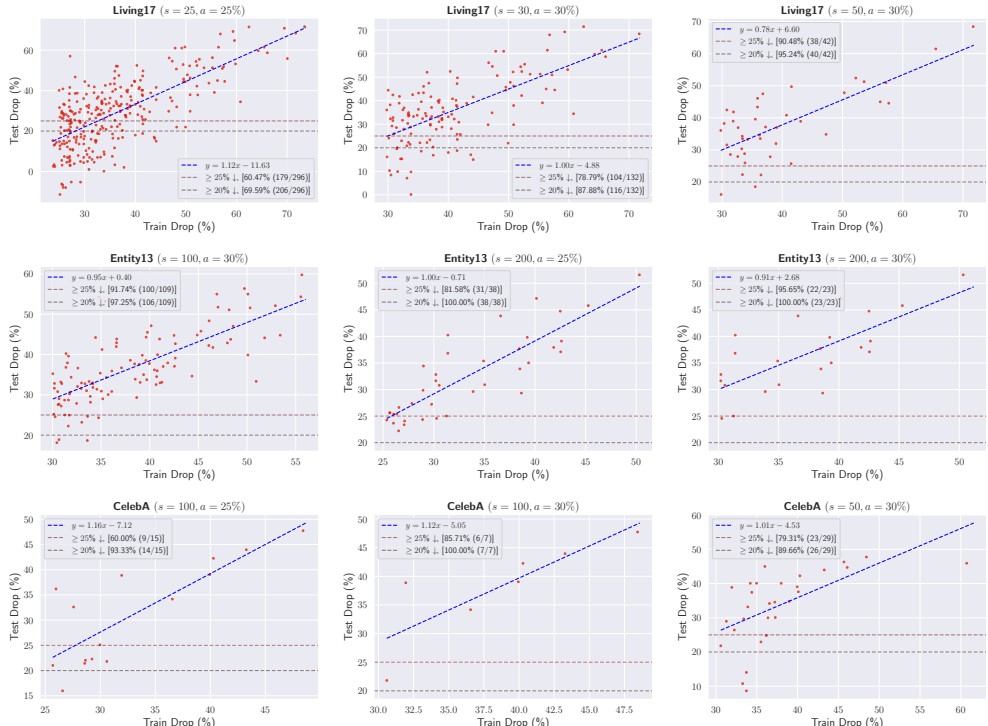

Figure 10: Evaluating detected failure modes on unseen (test) dataset. We extract failure modes on different datasets using different values of $(s, a)$.

However, we developed another greedy search algorithm where at each stage, we pick top tags that condition on them, significantly dropping model accuracy. By running this approach, the space of different choices shrinks and the algorithm becomes faster while missing some of the failure modes.

### A.7 DOMINO'S OUTPUT DESCRIPTIONS

To compare the results of DOMINO with `PRIME`, we picked DOMINO's hyperparameters in a way that generates relatively the same number of failure modes. Some of the outputs on Living17 dataset are as follows:

- class "salamander":
    - a photo of the bullet wound.
    - a photo of a lizard.
    - a photo of trout fishing.
    - a photo of a frog.
    - a photo of a hippo.
    - a photo of the ventral fin.
    - ...
- class "fox":
    - a photo of a gorillas.
    - a photo of the titanic sinking.
    - a photo of the tract.
    - a photo of a coyote.
    - a photo of oil shale.
    - a photo of the desert.
    - a photo of the antarctic.

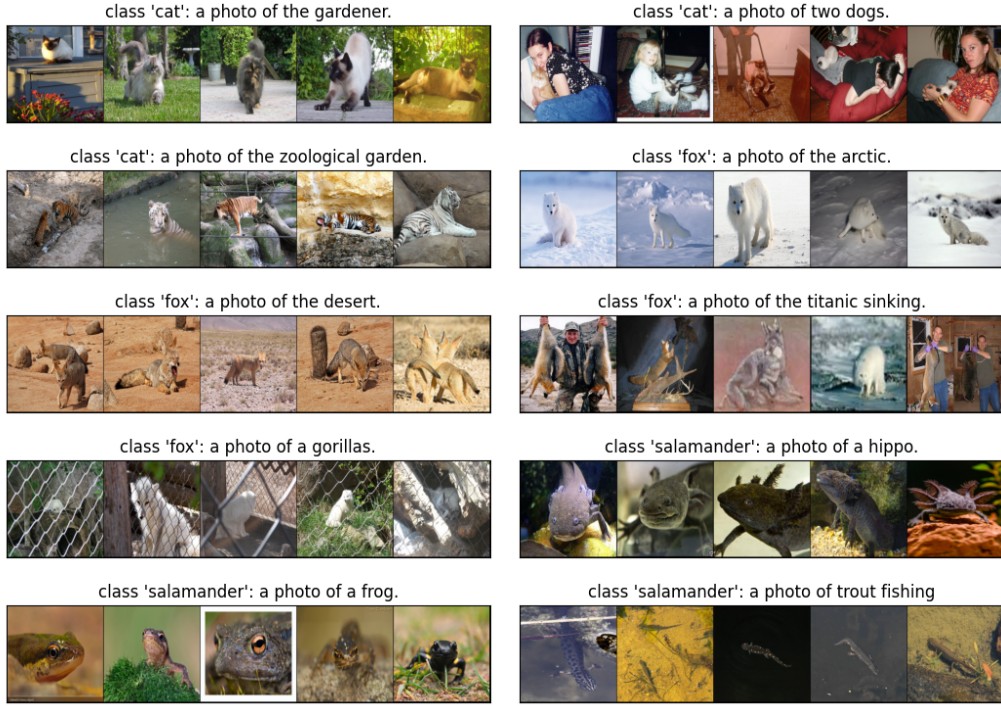

Figure 11: Some of the detected failure modes along with descriptions using DOMINO.

- a photo of the arctic.
- ...
- class "cat":
  - a photo of the zoological garden.
  - a photo of stray dogs.
  - a photo of two dogs.
  - a photo of the gardener.
  - a photo of the tehsil leader.
  - ...

Figure 11 shows some visualization of these failure modes. Lack of coherency among images in some of the groups as well as low-quality descriptions can be seen in detected failure modes.

## A.8   QUALITY OF DESCRIPTION

We note that failure modes detected by PRIME and DOMINO might be different. However, our metrics only evaluate how well failure modes are described with their corresponding captions and do not consider what images are assigned as failure modes. This enables us to compare different methods with each other. Notably, hyperparameters of different methods play a role in the number of failure modes that a method detects. In order to ensure a fair comparison, we carefully set the hyperparameters for both of methods to yield a similar number of detected failure modes.

Furthermore, it is worth noting that as the similarity score is normalized, comparing this score over different failure modes and different methods is possible. This is why we aggregate similarity score, standard deviation, auroc over all failure modes.

## A.9   DOMINO'S GENERALIZATION

To compare our results with DOMINO (Eyuboglu et al., 2022), we note that this method also outputs some groups as well as descriptions for them. For a failure mode $I_j$ with $T_j$ as its description, we

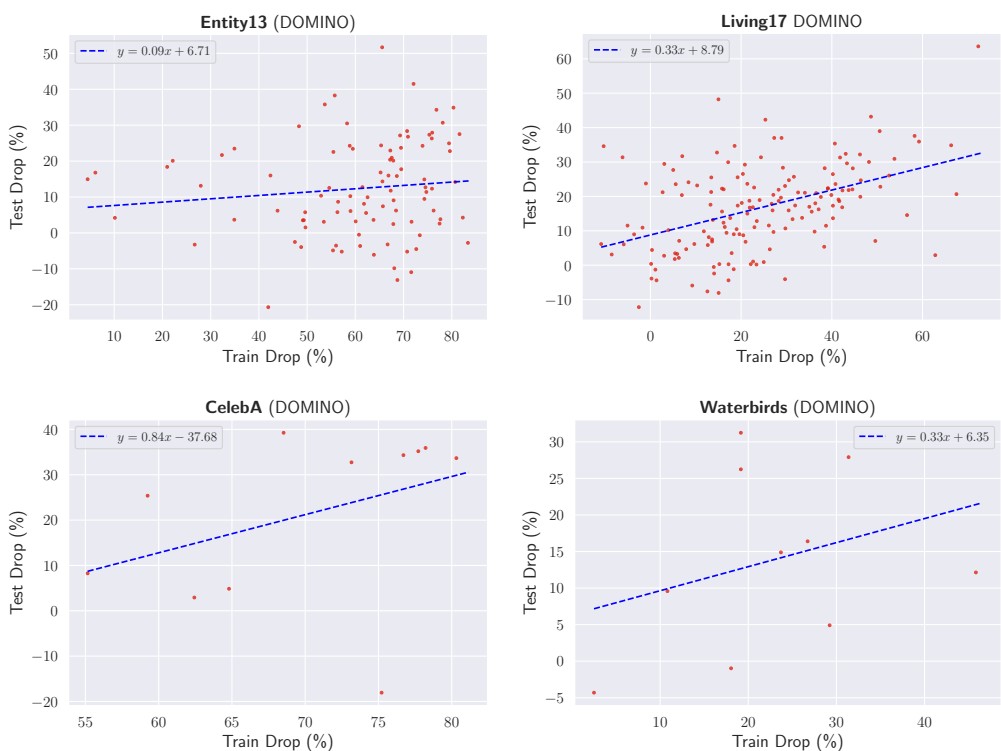

Figure 12: Evaluating detected failure modes by DOMINO on unseen (test) dataset.

use the same vision-language model that DOMINO uses to collect highly similar images to caption $T_j$ in $\mathcal{D}'$ and obtain $I'_j$. We then evaluate model's accuracy on images of $I'_j$ and expect a hard subpopulation, it should be as hard as $I_j$. Figure 12 shows the generalization of this method. We see a lower degree of generalization in this approach than our method. We observe that generated captions cannot fully describe as hard subpopulations as subpopulations detected on $\mathcal{D}$.

We note that the other method (Jain et al., 2022), only detects a single failure mode for each class in the input and reports around 10 images for that, thus, comparing this method with DOMINO and ours is a bit unfair as those methods detect multiple failure modes with significantly more coverage.

### A.10 IMAGE GENERATION

In this section, we elaborate more on the way we generate hard and easy images. To detect easy subpopulations, we randomly pick 2 subset of tags in a way that the model's accuracy on images representing those tags is $100\%$. For the failure modes, we randomly pick two of the detected failure modes in a way that those two groups do not share any common images. It is worth noting that for classes "butterfly" and "dog" we don't report any results as model's accuracy for these classes is almost $100\%$. Figure 13 shows the accuracy gap for different classes.

Here we provide some of the failure/success modes we took and corresponding descriptions used for generative models.

- class: "bear"
    - gray + water → "a photo of a gray bear in water";
    - river → "a photo of a bear in the river"
    - cub + climbing + tree → "a photo of a bear cub climbing a tree";
    - black + cub + branch → "a photo of a black bear cub on a tree branch";
- class: "ape"

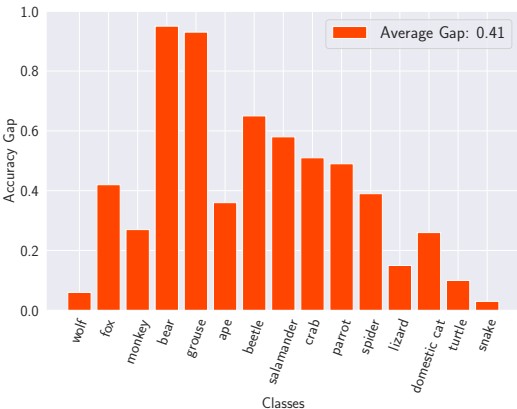

Figure 13: The difference in classifier accuracy between images generated from success mode and failure mode captions on Living17.

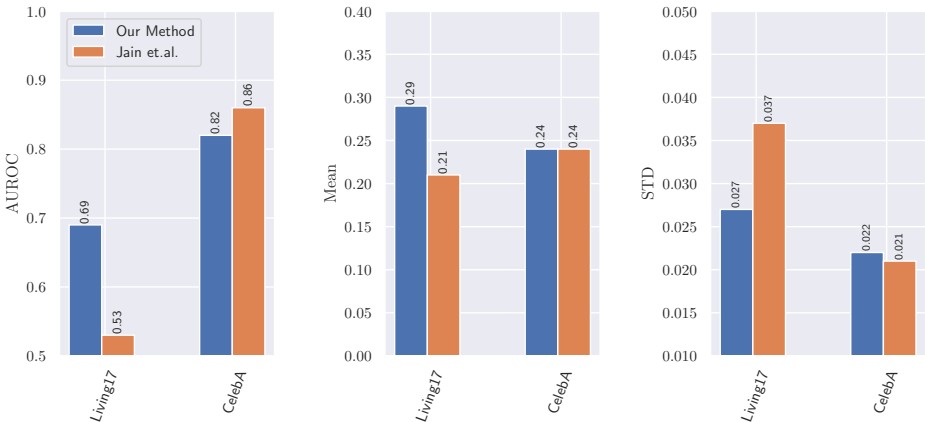

Figure 14: Comparison of the quality of description in our method and Jain et al. (2022). AUROC, Mean, and Standard over different datasets are reported.

- black + branch → "a photo of a black ape on a tree branch";
- sky → "a photo of an ape in the sky";
- gorilla + trunk + sitting → "a photo of a gorilla ape sitting near a tree trunk";
- mother → "a photo of a mother ape";

## A.11 JAIN ET AL. QUALITY OF DESCRIPTION

We note that Jain et al. (2022) detects 10 hard and 10 easy images for each class in the dataset and assigns a description to them. However, the way this method generates captions needs humans in the loop as they use a bunch of dataset-oriented words (tokens) that explain different variants of semantic attributes in the dataset. Hence, their method is not scalable to run on many different datasets so we only considered that on Living17 and CelebA. When datasets become larger and more complex, there will exist several failure modes, and Jain et al. (2022) that only extracts a single direction cannot cover many of the failure inputs. Figure 14 shows the results on different datasets. We note that on CelebA, they use a very detailed and manually collected set of tokens to generate outputs. However, even in that dataset, their performance is slightly better than ours.

Table 5: Accuracy on unseen (test) images for class ape when given tags appear in the inputs. More detailed descriptions not only bring more accurate explanation of images but also can lead to harder example where model's performance drops more severly. Some randomly sampled images within each group is visualized.

| Num of Tags | Tags | Sampled Images | Test Accuracy |
|---|---|---|---|
| 3 | hang; branch; black; |  | 41.18% |
| 2 | hang; branch; |  | 56.33% |
| | hang; black; |  | 56.25% |
| | branch; black; |  | 54.67% |
| 1 | hang; |  | 70.09% |
| | branch; |  | 69.23% |
| | black; |  | 73.21% |

Table 6: Average number of tags appear in at least $\alpha N$ images among $N$ closest images to a randomly sampled image in the representation space as well as average number of shared tags in semantic space (CelebA Dataset).

| | Representation Space | | | Semantic Space |
|---|---|---|---|---|
| | $\alpha = 0.6$ | $\alpha = 0.7$ | $\alpha = 0.8$ | |
| $N = 50$ | 4.17 | 3.56 | 2.91 | 7.55 |
| $N = 100$ | 3.94 | 3.34 | 2.71 | 7.25 |

## A.12 NUMBER OF TAGS IN THE DESCRIPTIONS

In Table 5, we bring a more detailed version of Table 1 that includes some sampled images of detected failure modes.

Table 7: Statistics of the distance between two points in CUB-200 dataset conditioned on number of shared tags. Distances are reported using CLIP ViT-B/16 representation space.

| # of shared tags $\geq d$ | mean | standard deviation | Probability |
|---|---|---|---|
| $d = 0$ | 7.37 | 0.92 | 0.50 |
| $d = 3$ | 7.31 | 0.94 | 0.48 |
| $d = 9$ | 7.06 | 1.08 | 0.42 |
| $d = 15$ | 6.29 | 1.88 | 0.30 |
| $d = 24$ | 3.42 | 3.18 | 0.12 |

Table 8: Average number of tags appear in at least $\alpha N$ images among $N$ closest images to a randomly sampled image in the representation space as well as average number of shared tags in semantic space (CUB-200 Dataset).

| | Representation Space | | Semantic Space |
|---|---|---|---|
| | $\alpha = 0.7$ | $\alpha = 0.8$ | |
| $N = 50$ | 2.53 | 1.46 | 12.34 |
| $N = 100$ | 3.00 | 1.99 | 10.91 |

## A.13 CUB-200 AND CELEBA

In this section, we show the results reported in Section 5.2 over CUB-200 dataset. Table 8 includes the results of shared tags in the proximity of images and Table 7 includes the statistics of distance between two images in the latent space.

## A.14 RAM EVALUATION ON LIVING17

we run a small-scale human validation study on Living17 (one of the main datasets we used in our paper) to evaluate the accuracy of RAM. We first obtain all tags of class "butterfly" in Living17 and filter out low-frequency tags as discussed in 3.1. 55 tags are remained, i.e.,

$$T_{\text{butterfly}} = \{\text{black}, \text{purple}, \text{red}, \text{flower}, \text{leaf}, \text{sit}, \text{land}, \text{gravel}, \text{stone}, \text{mud}, \text{wildflower}, \text{sky}, \text{grass}, ...\}.$$

We take 100 random images from the class "butterfly" in Living17 and evaluate precision/recall of tagging model on those images over tags of $T_{\text{butterfly}}$. Average Precision of RAM over those 100 images is $86.85\%$ and average recall is $81.85\%$. This shows that RAM is accurate in detection of tags in $T_{\text{butterfly}}$. It is worth noting that $T_{\text{butterfly}}$ includes a wide range of different objects and attributes, covering a wide range of concepts in those images.

## A.15 PRIME HYPERPARAMETERS

In this section, we elaborate more on each of the hyperparameters in our work. We note that all of the `PRIME`'s hyperparameters have intuitive definitions, enabling a user to calibrate `PRIME` toward their specific preferences.

- Parameter $a$ controls a trade-off between the difficulty and the quantity of detected failure modes. For example, selecting a high value of $a$ results in failure modes that are more difficult but fewer in number. Figure 10 shows this trade-off.

- Parameter $s$ determines the minimum number of required images inside a group to be detected as a failure mode. Small groups may not be reliable, thus, we filter them out. Larger value for $s$ results in more reliable and generalizable failure modes. The choice of $s$ also depends on the number of samples within the dataset. For larger datasets, we can assume that different subpopulations are sufficiently represented in the dataset, thus, a larger value for $s$ can be used. We refer to Figure 10 for observing the effect of $s$.

- $l$ determines the maximum number of tags we consider for combination. In datasets we considered, a combination of more than 5 tags would not result in groups with at least $s$ images, thus, we set $l \leq 4$ in our experiments. The choice of $l$ depends on the dataset and tagging model.

- $b_i$ refers to the degree of necessity for tags inside the failure modes. The current choice of $b_i$s is only a sample choice, requiring the appearance of each tag to have a significant impact on the difficulty of detected failure modes. One can adjust these hyperparameters based on their preference.

