# OpenReview forum: "PRIME: Prioritizing Interpretability in Failure Mode Extraction"
_ICLR.cc/2024/Conference — ICLR 2024 poster_

### Official Review · Reviewer_eU94 · 2023-10-30

**Soundness:** 3 good
**Presentation:** 3 good
**Contribution:** 3 good
**Rating:** 6
**Confidence:** 4

**Summary:**

The paper proposes a novel approach for identifying and describing failure modes in image classification models. The authors prioritize interpretability by analyzing the model's behavior based on the presence or absence of human-understandable concepts (tags) associated with images in the dataset. The method aims to generate high-quality text descriptions of failure modes while avoiding redundancy and noise. The paper presents experimental results on different datasets to demonstrate the effectiveness of the proposed approach.

**Strengths:**

1. Novel approach: The paper introduces a new method that prioritizes interpretability in failure mode extraction. By analyzing the presence or absence of human-understandable tags, the proposed approach aims to provide more accurate and meaningful descriptions of failure modes.
2. Experimental validation: The authors conduct experiments on different datasets to evaluate the effectiveness of their method. The results demonstrate that the proposed approach successfully identifies failure modes and generates high-quality text descriptions associated with them.
3. Importance of interpretability: The paper highlights the significance of prioritizing interpretability in understanding model failures. By providing human-understandable descriptions, the proposed approach enables easier identification and diagnosis of failure modes, leading to improved model reliability.

**Weaknesses:**

1. The authors mention comparisons with other studies (e.g., Eyuboglu et al., 2022; Jain et al., 2022) in the text but do not elaborate on the strengths and weaknesses of these studies and how this study compares favorably to them. It is suggested that the authors provide a more detailed explanation in this regard.
2. This paper mentions the impact of using different hyperparameter values on the results when discussing generalizability, but does not explicitly state the specific range and selection methods for these hyperparameter values. It is suggested that the authors provide detailed explanations in the paper regarding this so that readers can better understand and apply these methods.
3. Lack of comparison with state-of-the-art: The paper does not compare the proposed method with the current state-of-the-art methods for failure mode detection and description, making it difficult to assess its advancement. It is recommended that the authors perform a quantitative comparison with the latest methods.

**Questions:**

Please check the weakness.

---

> ### Author Response · Authors · 2023-11-18
>
> We appreciate the reviewer's interest in our work.
> We hope the reviewer finds our response helpful to reevaluate the paper and increase the score correspondingly.
>
>
> >  Comparison with an existing method
>
> 3rd paragraph In the introduction includes a brief comparison of the framework of our method and existing work. In section 4.3 and Appendix A.10, we compare our results with two prominent existing methods [1, 2] (state-of-the-art methods in failure mode detection and explanation with natural language). Our comparison evaluates the quality of generated captions. We will add a section to the Appendix covering more details about the way those methods work.
>
> The reason we state that our comparison is indeed in favor of existing methods is that they utilize CLIP to find text descriptions that maximize similarity score to images inside failure modes and we use CLIPScore (same metric) for comparison between different methods. Our method does not include CLIPScore in finding descriptions but we improve over existing work even in that metric.
>
> > This paper mentions the impact of using different hyperparameter values on the results when discussing generalizability
>
> Regarding the hyperparameters in our approach, we note that all of those hyperparameters have intuitive definitions, enabling a user to easily calibrate PRIME toward their specific preferences. We will add a more detailed section in the Appendix for these hyperparameters in the updated draft. Here are descriptions of hyperparameters in PRIME:
> + Parameter $a$ controls a tradeoff between the difficulty and the quantity of detected failure modes. For example, selecting a high value of $a$ results in failure modes that are more difficult but fewer in number. Figure 10 in the Appendix shows this tradeoff.
> + Parameter $s$ determines the minimum number of required images inside a group to be detected as a failure mode. Small groups may not be reliable, thus, we filter them out. Larger value for $s$ results in more reliable and generalizable failure modes. The choice of $s$ also depends on the number of samples within the dataset. For larger datasets, we can assume that different subpopulations are sufficiently represented in the dataset, thus, a larger value for $s$ can be used. We refer to Figure 10 in the Appendix for observing the effect of $s$.
> + $l$ determines the maximum number of tags we consider for combination. In datasets we considered, a combination of more than $5$ tags wouldn’t result in groups with at least $s$ images, thus, we set $l \leq 4$ in our experiments. The choice of $l$ depends on the dataset and tagging model.
> + $b_i$ refers to the degree of necessity for tags inside the failure mode. The current choice of $b_i$s is only a sample choice. We wanted the appearance of each tag to have a significant impact on the difficulty of detected failure modes.
>
>
> ---
> [1] Eyuboglu, Sabri, et al. "Domino: Discovering systematic errors with cross-modal embeddings." arXiv preprint arXiv:2203.14960 (2022).
>
> [2] Jain, Saachi, et al. "Distilling model failures as directions in latent space." arXiv preprint arXiv:2206.14754 (2022).

---

> > ### Author Response · Authors · 2023-11-22
> >
> > Hello, we hope our rebuttal has addressed your comments. Please let us know if you have any additional questions, we would be more than happy to answer. Thank you for your time and feedback.

---

### Official Review · Reviewer_j2ri · 2023-11-01

**Soundness:** 3 good
**Presentation:** 3 good
**Contribution:** 4 excellent
**Rating:** 8
**Confidence:** 3

**Summary:**

This paper addresses the problem of understanding misclassifications in image classifiers. It proposes an approach called PRIME, that uses a set of tags that correspond to human understandable concepts, and finds what combinations of tags align (using a vision-language model) with misclassified subgroups of images. This contrasts with prior work that first clusters misclassified images and then attempts to label them with human understandable concepts, but suffers from the problem that similarity in feature space might not necessarily imply similar features in the image, leading to noisy concept descriptions. A comprehensive experimental evaluation is performed to demonstrate the effectiveness of the proposed approach.

**Strengths:**

1. The paper addresses the important problem of understanding failure modes in deep networks. Given the black-box nature of such models, this is crucial building trust and deploying such models safely.
1. It recognizes and addresses a drawback in prior clustering based approaches, that similarity in representation space need not imply similarity in semantic space. This corroborates findings reported in other contexts in prior work (e.g. [1]). It proposes a simple alternative, i.e. tagging all images and finding subgroups that perform poorly, and shows that it is much more effective.
1. By labelling tags first, PRIME finds exactly what combinations of tags cause the model to fail, instead of just finding what is common in failure cases. The latter may not answer whether every tag in the set is necessary, and so the set may not be minimal. This represents more faithful explanations for the failure modes.
1. The evaluation performed is comprehensive and across a variety of dimensions, including generalization to generated images, unseen images, and for the quality of the tags. The evaluation also shows the effectiveness of PRIME as compared to prior work.
1. The paper is well-organized and easy to follow.


[1] Hoffmann et al. This Looks Like That... Does it? Shortcomings of Latent Space Prototype Interpretability in Deep Networks. ICML-W 2021.

**Weaknesses:**

1. The method seems highly reliant on being able to find all relevant tags first. Failure modes caused by concepts not in the tag set $T_c$ would remain undetected. Given that low frequency tags are filtered out, this could miss potentially important but less frequently occurring failure modes, such as spurious correlations. Broadly, there seems to be a tradeoff involved in choosing the size of the tag set -- larger the set, more the failure modes that can be caught, but also more computationally expensive the method is.
1. The effectiveness of the method also depends on ensuring that a tag for a concept is actually matched exactly to images containing the concept (i.e., to the accuracy of the Recognize Anything Model). An evaluation of whether this actually happens would have been helpful, especially with datasets that have attribute labels that could potentially be matched to generated tags.
1. The results in Figure 4 and other similar plots could be more rigorously shown by reporting the correlation between the accuracy drop in the training and test sets for each subgroup.

**Questions:**

Discussion on the points raised in Weaknesses would be helpful.

---

> ### Author Response · Authors · 2023-11-18
>
> We are glad that the reviewer finds our work interesting.
>
> > The method seems highly reliant on being able to find all relevant tags first. Failure modes caused by concepts not in the tag set would remain undetected.
>
> We first refer the reviewer to G1 for tags inside $T_c$. Rare tags are removed to avoid any noisy or irrelevant ones in the algorithm. We agree with the reviewer that this can lead to missing some specific hard subpopulations but low-frequency tags represent subpopulations that do not include enough samples and in PRIME we ignore small subpopulations because they might not be reliable.
>
>
> > The effectiveness of the method also depends on ensuring that ...
>
> We refer the reviewer to G1.
>
>
> > The results in Figure 4 and other similar plots could be more rigorously shown by reporting the correlation between the accuracy drop in the training and test sets for each subgroup.
>
> Thanks for your suggestion. We’ll add correlations to the updated plots.
> Here are the correlations for the plots in Figure 4, 10:
>
> **Living17**
>
> | $s$      | $a$ | corrcoef |
>  | ----------- | ----------- | ----------- |
> | 25 | 25 | 0.70 |
> | 50 | 30 | 0.73 |
> | 30 | 30 | 0.72 |
>
> ---
>
> **Entity13**
>
> | $s$      | $a$ | corrcoef |
>  | ----------- | ----------- | ----------- |
> | 100 | 30 | 0.73 |
> | 200 | 30 | 0.73 |
> | 200 | 25 | 0.85 |
>
> ---
>
> **CelebA**
>
> | $s$      | $a$ | corrcoef |
>  | ----------- | ----------- | ----------- |
> | 50 | 30 | 0.60 |
> | 100 | 30 | 0.83 |
> | 100 | 25 | 0.81 |

---

> > ### Comment · Reviewer_j2ri · 2023-11-23
> >
> > Thank you very much for the response and the additional analyses. I agree that this work is interesting and useful and I would like to keep my rating unchanged.

---

### Official Review · Reviewer_dBy1 · 2023-11-02

**Soundness:** 3 good
**Presentation:** 3 good
**Contribution:** 3 good
**Rating:** 6
**Confidence:** 4

**Summary:**

The authors introduce PRIME, a new method to identify (text descriptions of) coherent groups where a trained image classifier underperforms. PRIME works by (1) "tagging" each image using a pretrained model, and (2) performing an exhaustive search over the sets defined by all possible combinations of tags, for the sets which are "minimal", larger than some threshold, and have accuracy lower than some threshold. Like past methods, PRIME is "unsupervised" in that it does not need a HIL or a comprehensive existing set of tags.


The authors evaluate PRIME in Section 4 by reporting the "generalization" of the generated text descriptions (when used to retrieve matching unseen images from a test set or new generated images), and propose three new "metrics" that compare the CLIP embeddings of the text description to the images in each group.  They also run additional experiments in Section 5 that further support their claims by demonstrating the "value" of defining more specific subgroups using combinations of tags, and also exploring properties of pretrained embeddings such as CLIP.

**Strengths:**

1. The authors contextualize related work on error discovery well in the Introduction, and clearly motivate gaps in the existing literature (i.e., that generating text descriptions of groups where an image classifier underperforms is difficult). This problem is timely and significant.
2. The authors' instructive figures (e.g., Figures 1, 3, and 5) clearly illustrate the benefits of their proposed approach.
3. The authors' proposed PRIME method is straightforward, and explained in a way that is clear and reproducible as a reader. I specifically like how the authors motivated why they are interested in searching for a minimal set – to my knowledge, this criteria, while intuitive, has never been discussed in the context of error discovery.
4. I appreciated the author's exploration of potential limitations with clustering-based approaches by running experiments using CLIP in Section 5.2.  Given the popularity of such embeddings in the field of error discovery, I thought this analysis was particularly timely and significant.

**Weaknesses:**

My primary critique of this work is that I believe the authors should directly address potential limitations of their proposed method in their paper.  In the present draft, many important limitations are excluded completely.  I am willing to adjust my score if my below concerns are addressed.

* **Weakness #1: Understanding limitations of relying on a pre-trained tagging model (Step 1)**. PRIME relies on a pre-trained tagging model (in this case, RAM) to provide a set of tags for each image. However, relying on a pre-trained model may have several limitations:
  * As a simple example, when we consider the "fox+white+zoo" group in Figure 1, it is completely possible that RAM may fail to tag a large number of images in the dataset that are in a "zoo" (false negatives), or include images that are not taken in a zoo in the group (false positives), etc.
  * The tags output by RAM also are likely not comprehensive – there are some objects that perhaps RAM is worse at recognizing due to their absence in the pretraining data.
  * I may be missing something but the present draft is also unclear on how many tags are sampled for each image in this first part, and how the tags are sampled (do you only look at the 100 "most likely" tags for each image?).
  * All of the authors' evaluation experiments in Section 5 appear to assume that the tags generated by RAM identify groups of images that "match" them; however, without further evidence I believe that this assumption is unfounded. I would be interested in whether the authors can perform any kind of small-scale evaluations where they report the precision/recall of the RAM tagger – perhaps you can even compare its performance at tagging some classes that you already have metadata labels available for.
* **Weakness #2: Clarify how the "tags" should be interpreted (moving from tags to descriptive captions)**.
  * Each group is defined as the intersection of a set of tags that are all present in that group's images. But, some ambiguity remains in how to interpret the tags.  While the example groups that the authors selected in Figures 1 and 2 have clear and natural interpretations, in general I think the ambiguity of modifiers, attributes, and propositions (relationships between multiple objects) may be difficult to interpret.  Even for the group "fox + white" (which the authors interpret as foxes with white fur), it's unclear to me if this is a correct interpretation because the tagging model may include any photo that has "white" anywhere at all in it (ex, an orange fox in the snow).  I believe that leaving this point unaddressed may mislead users of PRIME to draw the wrong conclusions.

* **Weakness #3:  Clarify experimental methodology.**  I would like to see more detail in the main text about some parts of the experimental methodology to ensure that the examples provided in the paper are indeed representative of PRIME's performance (instead of being cherry-picked to make PRIME look good).
  * Specifically, how exactly did you "use ChatGPT" to move from sets of tags, to semantically meaningful captions, in Section 4.2 (and did you do any qualitative verification that these generated captions indeed seemed to "match" their corresponding groups of images)?
  * How did you select the images that you're visualizing for each of the groups in Figures 1, 3, and 5 – were they randomly selected, or chosen (related to my Weakness #1)?

**Questions:**

See Weaknesses above.

In addition my listed weaknesses (which are more important to address), I had a few remaining clarifying questions/suggestions:
* Make clear in the introduction that your method doesn't rely on human-labeled tags, but rather uses a pre-trained model
* nit (p2): "images who match group of tags identified as a failure mode" => "the images that match a failure mode"...
* nit (Section 4): rather than saying "difficulty (of the detected failure modes)", explicitly use terms like "accuracy" or "performance gap", as "difficulty" is vague
* Section 5.2: I want to see more nuance in this discussion about how it seems like your method will always incentivize choosing a higher-error subclass (rather than a larger superclass), when in reality it may be of interest to return the superclass.  I don't think that "smaller"/"more specific" is always better in practice.  I think what's interesting about the exhaustive search here is that you can present the performance over all of the possible subgroups to the model developer/user, who can then choose which ones they would like to prioritize for fixing or further evaluation moving forward.
* Two helpful references to consider adding to your related work section that illustrate the difficulty of generating high-quality text descriptions for error discovery are (1) Appendix C of AdaVision [1], and (2) the user study in Johnson et al. [2],

[1] https://arxiv.org/abs/2212.02774
[2] https://arxiv.org/abs/2306.08167

---

> ### Author Response · Authors · 2023-11-18
>
> We appreciate the reviewer for accurate points about our work.
> We hope the reviewer finds our response helpful to reevaluate the paper and increase the score correspondingly.
>
> ### Understanding limitations of relying on a pre-trained tagging model
> > As a simple example, when we consider the "fox+white+zoo" group in Figure 1 ...
>
> We first refer the reviewer to G1, where we affirm RAM's reliability via large scale evaluations from their original paper and a small scale human study we conducted ourselves validating RAM's performance on the datasets we used.
>
> Secondly, it is worth noting that other state-of-the-art methods also suffer from the same issues, e.g., it is possible that some images with the same semantic attributes are not assigned to the detected clusters (false negatives), or some samples that might be different from other images in the cluster are assigned to that (false positives). Furthermore, this also happens when finding the text that maximizes similarity to images inside the group, the text may not cover all common attributes within images inside the cluster while covering some attributes that might be present in images outside of the failure mode.
>
> This is why we measure similarity, specificity, and coherency in Section 4.3 across different methods and observe improvements in our method. This shows that PRIME, even though it is not perfect and has these limitations as the reviewer pointed out, can be more promising.
>
> > The tags output by RAM also are likely not comprehensive ...
>
> We refer the reviewer to G1.
>
> > I may be missing something but the present draft is also unclear on how many tags ...
>
> We use the default set of hyperparameters for RAM that extracts tags from images. Number of tags that RAM generates is not fixed and depends on the image. Number of tags range from 5 to 20.
>
> > the authors can perform any kind of small-scale evaluations where they report the precision/recall of the RAM tagger.
>
> We refer the reviewer to G1.
>
> ### Clarify how the "tags" should be interpreted (moving from tags to descriptive captions)
> We appreciate the reviewer for pointing out this limitation. Interpreting some of the tags (especially attributes) might be a bit misleading. We will amend the text to avoid the confusion that the tag “white” in fox class means “fox with white fur”. In fact, our results show that at the intersection of some tags (not always related to the main object in the image), there might be the model’s failures.
>
>
> ### Clarify experimental methodology
> > Specifically, how exactly did you "use ChatGPT" to move from sets of tags ...
>
> We refer the reviewer to G2.
>
> > How did you select the images that you're visualizing for each of the groups in Figures 1, 3, and 5 ...
>
> Images are randomly selected from the group of images representing those tags. This is the case not only for those Figures but also other additional images in the Appendix.
>
> > Make clear in the introduction that your method doesn't rely on human-labeled tags, but rather uses a pre-trained model,
>
> Sure, we’ll explicitly mention that in the updated draft.
>
> > nit (Section 4): rather than saying "difficulty (of the detected failure modes)", explicitly use terms like "accuracy" or "performance gap", as "difficulty" is vague, nit (p2): "images who match group of tags identified as a failure mode" => "the images that match a failure mode"...
>
> Thanks for pointing this out, we will fix this issue in the updated draft.
>
> > method will always incentivize choosing a higher-error subclass …
>
> We agree with the review that superclasses are more interesting. This is exactly why we added more constraints for reporting more specific subclasses that are represented by more tags. In fact, when we report a failure mode with a greater number of tags (more specific), we ensure the necessity of all these tags. Removing any of those tags will result in easier subpopulations. In summary, superclasses that are hard enough are reported as well as more specific subpopulations that are hard enough and also harder than their superclasses. As mentioned by the reviewer, exhaustive search allows us to consider all possibilities and enables users to pick failure modes they want.
>
> > Two helpful references to consider adding to your related work section …
>
> Thanks for pointing out these references, they are indeed relevant and aligned with our point of view in this paper. we will add them to the updated draft.

---

> > ### Comment · Reviewer_dBy1 · 2023-11-19
> > **Response to rebuttal**
> >
> > Thank you for your thoughtful response to my review.
> >
> > * **re: Weakness #1**, I appreciate the authors' G1 response, and agree with their point that most existing methods rely on some auxiliary model.  I do not think that doing so is a weakness of this work; I rather just wanted the authors to explicitly acknowledge that this model may be imperfect and make some effort to interrogate and measure the capabilities and limitations of RAM (which they claim is current SOTA).  To not do so would be irresponsible given that the authors hope that practitioners may use (and perhaps even over-rely) on PRIME to debug their models in the future. I appreciate that the authors ran an additional human subject study.  From their description in the general response, however, I still have several questions about the study.  For which "tags" did you validate using human subjects -- did you use the same tagging procedure as your paper?  Did you limit the space of tags that were sampled in this experiment? How did you define the "false negative" tags that were missed by RAM (or did you only check if the tags generated by RAM were actually in the image)?
> > * **re: Weakness #2**,  I found your G2 response to be clarifying (I hope you include this information in your updated manuscript), but it still doesn't fully address my concern.  I believe not only that interpreting the set of tags is difficult, but also that there may be some implicit limitations to defining each bug as the intersection of a set of tags.  For example, consider "fox+white+snow".  A simple object detection/tagging model would label both images with a white fox, or an orange fox in white snow, as belonging to this group.  e.g., there would be no way to differentiate "white fox" vs. "white snow" using this definition.  In general again as I mentioned earlier this definition may fail to attribute the appropriate adjectives/qualifiers to the right objects in a scenario where there's multiple objects being used to define the bug.  I think this limitation is OK - but should be acknowledged!
> >
> > In summary, I appreciate the authors for engaging with my review, but still have several remaining concerns with the present version of the draft. I also understand that we are at the end of the discussion period, but I would appreciate if the authors could have updated their manuscript with the full details of the requested clarifications and experiments (rather than making commitments or summarizing new experiments at a high level).  For these reasons my score remains unchanged.

---

> > > ### Author Response · Authors · 2023-11-20
> > >
> > > We thank the reviewer for prompt response.
> > > We have revised the draft, and a fresh version has been uploaded. We hope that the reviewer finds the new version noteworthy, potentially prompting a reevaluation that could positively impact the score.
> > >
> > > > however, I still have several questions about the study. For which "tags" did you validate using human subjects -- did you use the same tagging procedure as your paper? Did you limit the space of tags that were sampled in this experiment? How did you define the "false negative" tags that were missed by RAM (or did you only check if the tags generated by RAM were actually in the image)?
> > >
> > > We took $T_{\\text{butterfly}}$ including tags that RAM extracts from images of butterfly class in Living17 after removing low-frequency tags with exactly the same procedure we used in PRIME. This set includes $55$ tags covering wide range of objects, attributes, colors, etc.
> > >
> > > $T_{\\text{butterfly}} = \\{\\text{black}, \\text{purple}, \\text{red}, \\text{flower}, \\text{leaf}, \\text{sit}, \\text{land}, \\text{gravel}, \\text{stone}, \\text{mud}, \\text{wildflower}, \\text{sky}, \\text{grass}, ...\\}.$
> > >
> > > We randomly took $100$ images of this class and obtained false negatives by looking at tags inside the set that are clearly presented in the image but not detected by tagger, and false positives by looking at the tags detected by the model but not clearly presented in the image. Two people did this experiment and we reported the average of those metrics. we used a small sample of images due to the time constraint of the rebuttal period.
> > >
> > > > I found your G2 response to be clarifying (I hope you include this information in your updated manuscript), but it still doesn't fully address my concern ...
> > >
> > > We appreciate the reviewer for pointing this out. In the new version, we added this discussion that interpreting some of the tags is challenging as they may not correspond to the main object in the image. We have also amended the text to avoid any misunderstanding.

---

> > > > ### Comment · Reviewer_dBy1 · 2023-11-20
> > > > **Response to updated draft**
> > > >
> > > > Thank you for addressing my additional questions, and also for updating the draft to include the additional experiments and a limitation section.  After reviewing the modified draft, I believe that the present draft is significantly better in its clarity and acknowledgment of limitations.  However, I still do believe that the authors' proposed method still has several limitations that I discussed above, such as the under-specificity of a set of tags to define bugs. As such I have raised my score to recommend weak acceptance.  Thank you for engaging thoughtfully throughout the rebuttal.

---

### Official Review · Reviewer_WEiw · 2023-11-09

**Soundness:** 1 poor
**Presentation:** 2 fair
**Contribution:** 2 fair
**Rating:** 5
**Confidence:** 4

**Summary:**

This paper proposes a method to find human-understandable descriptions of situations under which a given image classification model fails. The authors argue that prior works obtain these descriptions using clustering in latent spaces which results in poor identification of shared attributes. In their method, they first obtain _tags_ for images using a SOTA model, and then find combinations of tags for which the model performs poorly to understand failure modes. They claim that this _interpretability-first_ approach results in a more robust failure mode extraction method.

**Strengths:**

1. Experiments done on multiple datasets to support the approach.
2. Visual examples of the method working look promising. The proposed method maybe a good practical method to generate descriptions of failure modes in a general purpose entity recognition models.

**Weaknesses:**

1. The method relies heavily on an auxiliary model (like Recognize Anything Model). Although this makes an important tool for describing failure modes, everything that the proposed method can do is bottlenecked by this auxiliary model's capability. For example, the authors present this method as a tool to describe "failure modes in trained image classification models". However, what happens when the image classification model is trained on a domain-specific dataset like chest X-Rays? How can we expect the auxiliary general purpose model to provide tags for this dataset? This is a practical paper but we do not see ablation studies with other general purpose auxiliary models (apart from RAM), or other experiments with domain-specific image classification models.
2. The tag generation process requires a threshold hyper-parameter. Similarly, there a lot of hyper-parameters in the proposed method's setup like $b_i$, $a$, $s$, $l$, etc. Some of them look more sensitive than others. However, no insights are given on how they are selected apart from saying "we generally" use these values.
3. In 4.1 paragraph 1, the authors say that $D$ and $D'$ are from the same distribution, and hence the accuracy drop should translate. This is not what generalization means. Generalization applies to the scenario when unseen data is from a shifted distribution. As the authors clearly mention that the unseen dataset is drawn from the same distribution (and not explicitly curated in some way like selective sampling), this dismisses all claims of generalization. The generalization on generated data is an interesting experiment but the fact that authors fine-tuned the generated model on the training data simply nullifies all claims of generalization.
4. The authors say, "We utilize language models to create descriptive captions for objects and tags associated with failure modes in images". It is unclear how this is done. And how is the quality of the generated captions measured?
5. Regarding the quality of descriptions metrics: Firstly, I have no idea what the AUROC metric means at all. Moreover, these "three" metrics are again based on another auxiliary model. It suffers from all the same issues as mentioned in point 1. The authors don't talk about this potential pitfall. Additionally, this evaluation goes through multiple models to come up with embeddings to calculate a distance metric. First, the tags are generated using RAM. After the failure mode tags are identified, they are passed through a language model to generate captions. Then, the caption is passed through another foundation model to generate text embeddings while the original image is passed through the same foundation model to generate image embeddings. I don't see how we could make any definitive claim about these metrics (which don't seem to be very different from the corresponding values of DOMINO).
6. Regarding the similarity between representation space and semantic space: This analysis is confusing and potentially flawed. The representation space may carry nuanced information which is not captured by the tags. For example, a common tag which says "water" may have very different representations depending on whether it is ocean blue water or river green water. Therefore, it can not be expected that closeness in representation space should closely align with closeness in semantic space. The semantic space is formed after (potentially several) layers of abstraction on the representation space which is supposed to store all fine-grained details. The fact that table 5 still shows a significant number of overlapping tags in nearby images is something that I would expect to begin with. The images that share maximum number of tags might have very different subtle differences that make them distant in the representation space.

**Questions:**

See weaknesses.

Overall, I feel that although the presented approach may yield good failure mode descriptions for some general purpose models, the study is not sound enough. Moreover, the paper lacks a section on limitations (in my view, there are many). All this fails to excite me to support of the paper for acceptance.

---

> ### Author Response · Authors · 2023-11-18
>
> We thank the reviewer for accurate points about the paper. We hope that this rebuttal can address the concerns and result in higher scores for this work.
>
> > The method relies heavily on an auxiliary model … can be bottlenecked by this auxiliary model's capability.
>
> We refer the reviewer to G1.
>
> > what happens when the image classification model is trained on a domain-specific dataset like chest X-Rays?
>
> As mentioned in G1, tagging models will be developed for specific domains. As a result, PRIME’s framework can be applied in those domains as well. We note that this is the case in existing work where a fine-tuned version of CLIP on specific domains is needed for those methods to be effective. We appreciate the reviewer for pointing this out, we will add this discussion to the updated draft.
>
>
> > The tag generation process requires a threshold hyperparameter. Similarly, there are a lot of hyper-parameters in the proposed method's setup like $s$, $a$, ...
>
> We used the default hyperparameters of RAM. Regarding the hyperparameters in our approach, we note that all of those hyperparameters have intuitive definitions, enabling a user to easily calibrate PRIME toward their specific preferences. We will add a more detailed section in the Appendix for these hyperparameters in the updated draft. Here are descriptions of hyperparameters in PRIME:
> + Parameter $a$ controls a tradeoff between the difficulty and the quantity of detected failure modes. For example, selecting a high value of $a$ results in failure modes that are more difficult but fewer in number. Figure 10 in the Appendix shows this tradeoff.
> + Parameter $s$ determines the minimum number of required images inside a group to be detected as a failure mode. Small groups may not be reliable, thus, we filter them out. Larger value for $s$ results in more reliable and generalizable failure modes. The choice of $s$ also depends on the number of samples within the dataset. For larger datasets, we can assume that different subpopulations are sufficiently represented in the dataset, thus, a larger value for $s$ can be used. We refer to Figure 10 in the Appendix for observing the effect of $s$.
> + $l$ determines the maximum number of tags we consider for combination. In datasets we considered, a combination of more than $5$ tags wouldn’t result in groups with at least $s$ images, thus, we set $l \leq 4$ in our experiments. The choice of $l$ depends on the dataset and tagging model.
> + $b_i$ refers to the degree of necessity for tags inside the failure mode. The current choice of $b_i$s is only a sample choice. We wanted the appearance of each tag to have a significant impact on the difficulty of detected failure modes.
>
>
> > the authors say that  $D$ and $D’$ are from the same distribution …
>
> We note that in failure mode extraction, we do not intend to consider the performance of the classifier on out-of-distribution datasets. In principle, “generalization” refers to the model's ability to adapt properly to new, previously unseen data, drawn from the same distribution as the one used to create the model. By generalization, we showed that failure modes we extract from training data can generalize on unseen (test) data. In other words, hard subpopulations that are extracted from training data, represent hard subpopulations on test data. This measures the reliability of PRIME.
>
> Regarding the experiments about image generation, we want to emphasize that we need to finetune the model on the Living17 dataset to generate images that come from the distribution that the classifier can classify. Otherwise, if generated images are out of distribution, the classifier does not have a good performance for any generated image, making the drop in accuracy for generated images from a hard supbopulation vacuous. Now, we do observe that the classifier performs well on easy subpopulations and loses its accuracy over harder ones.
>
> > The authors say, "We utilize language models to create descriptive captions …
>
> We refer the reviewer to G2.
>
>
> >  Moreover, these "three" metrics are again based on another auxiliary model.
>
> For evaluating the quality of descriptions, as discussed in the last paragraph of the introduction, it is hard to scalably evaluate, thus, we need to utilize vision-language models (CLIPScore) as a proxy to evaluate them. This is standard practice as other researchers also use foundation models (CLIPScore, BertScore, etc) for large-scale evaluations.
>
> Furthermore, existing methods are *designed to maximize CLIP similarity* of the description of a failure mode to images inside that. We used this metric to show the advantage of our method, as we improve over existing results in similarity, specificity, and coherency despite the fact previous methods *optimize similarity metric by design*.

---

> > ### Author Response · Authors · 2023-11-18
> >
> > > Firstly, I have no idea what the AUROC metric means at all.
> >
> > AUROC measures the specificity of descriptions to images inside the failure modes. For each failure mode, we take the similarity score of the description and images inside that. We also sample some images outside of the failure mode and take the similarity score to the description. Images outside should have smaller similarities than images inside the failure mode. This is why we report AUROC between these two groups to measure the specificity. As mentioned in G2, we generate descriptions in a bag-of-word manner, not with language models.
> >
> > > which don't seem to be very different from the corresponding values of DOMINO
> >
> > We note that we have improved mainly in specificity and coherency while slightly improving in similarity score. This comes even though DOMINO uses CLIP and finds text that maximizes this similarity.
> >
> > > Regarding the similarity between representation space and semantic space
> >
> > We agree with the reviewer that representation space is indeed practical, useful, and informative. We agree that representation space is not equivalent to semantic space. However, when we need to *assign human-understandable descriptions* to failure modes by considering a cluster in representation space, there must be semantic similarities between those images in a way that we can describe them with *language tags and tokens*. As a result, if images inside a cluster in proximity in the latent space do not share any common semantic attribute, it is hard to describe them in natural language which is the goal of this line of research.
> >
> > We refer to Figure 11 for some visualization of images that are within a cluster detected by DOMINO but they may not share semantic (human-understandable) concepts, thus, making it harder to describe those clusters with human language. We wanted to highlight the point that the quality of description degrades for clustering-based approaches, hence, we propose another direction that sounds promising.
> >
> > We thank the reviewer for all the detailed points. We will include all the above limitations in a new section for the updated version of our paper.

---

> > > ### Comment · Reviewer_WEiw · 2023-11-21
> > > **Response to Rebuttal**
> > >
> > > I apologize for the late response. I have been busy with rebuttals of my papers as well.
> > >
> > > > As mentioned in G1, tagging models will be developed for specific domains. As a result, PRIME’s framework can be applied in those domains as well.
> > >
> > > It is unreasonable to assume that tagging models can be developed for specific domains where the both data and annotators are scarce. The main concern here is the significance of the contribution and claims. Heavy-lifting is done by the tagging model, so the "framework" of PRIME, which just finds common tags in failing samples (by brute force search), does not seem to be very novel. This could be done in multiple different ways like clustering, etc. Moreover, I see that the updated draft's abstract still over-claims the proposed methodology as a providing human-understandable descriptions for failure modes in **trained image classification models** which encompasses all classification models and not just ones for which RAM finds tags.
> > >
> > > > In principle, “generalization” refers to the model's ability to adapt properly to new, previously unseen data, drawn from the same distribution as the one used to create the model.
> > >
> > > Although I agree that generalization is also used in this sense, to show that the model has not overfit on training samples, it is not of much interest in cases where ample images were used to get generalizable failure modes. The interesting parameter here is the number of samples in $D$. I see one dataset size (88400 of Living17) in the paper which kind of makes my point. In my opinion, randomly sampling even half of these many images for generating failure modes, would essentially cover the distribution quite well and therefore the generalization result (on the other half) would be very much expected. The generalization claim here would be much more interesting if the authors generated failure modes on small subsets of the dataset on which the tagging approach might overfit and still showing generalization.
> > >
> > > > Regarding the experiments about image generation
> > >
> > > I agree with the authors that if generated images are OOD in the sense that if the classifier was trained for detecting cat vs dog and you showed it an image of a rocket in space, the failure modes would probably not make much sense. But my issue was with fine-tuning the generative model on the same dataset. The authors could have used a conditional generative model to generate images that have the same "class" like images of dogs, but not necessarily the semantic details that are similar to the training set. This would be a more interesting study as it would measure how well the given classifier's understanding aligns with an unrelated generative model's (in terms of failure to understand and generate failing features).
> > >
> > > > For evaluating the quality of description ... This is standard practice ...
> > >
> > > I am not familiar with the literature in failure mode detection, but using an artificial metric that goes through multiple models' embedding spaces is seriously problematic as it assumes their representation of the world to be perfect. There is no way to know how reliable this is for a particular image.
> > >
> > > > ... if images inside a cluster in proximity in the latent space do not share any common semantic attribute, it is hard to describe them in natural language ...
> > >
> > > This is exactly my point regarding the reliance on auxiliary models. This will happen when the representation (embedding) generated by the auxiliary model is non-representative of all features. Deep models sometime make uninterpretable mistakes when inputs lie in the tail of distributions. So there will be instances when there are common mistakes of the same kind but we as humans can't interpret them. These are the actual situations when we need help of such methods to tell us what's going wrong.
> > >
> > > In summary:
> > > The authors have addressed some questions of mine, but my primary concerns with the work remain. I see very little original contribution in the work. It mostly relies on using other works' methods and combine them to give failure mode detection. The evaluation regarding generalization and quality of descriptions seem very superficial. Moreover, there are several limitations of the work which are still deferred to the appendix and I think that should be in the main text.
> > > If there was an option to raise the score to 4, I would do that. I am raising my score to 5 just to motivate the authors to make more holistic efforts in studies of this kind, but I will still reject this paper for publication.

---

> > > > ### Author Response · Authors · 2023-11-22
> > > >
> > > > We greatly thank the reviewer for reevaluating our work and improving the score. Here are some comments regarding your concerns.
> > > >
> > > > > It is unreasonable to assume that tagging models can be developed for specific domains where the both data and annotators are scarce.
> > > >
> > > > We note that as other vision-language models (such as CLIP) have been  developed for specific domains, e.g., ConVIRT (CLIP for medical data), tagging models can also be developed for them. For example, one can finetune RAM with image-caption datasets from specific domains. Overall, we propose PRIME as a general framework for detecting and explaining failure modes and we acknowledge that its performance is affected by the auxiliary model used for tagging. More advanced tagging model can leads to improvement in PRIME’s performance.
> > > >
> > > > > So there will be instances when there are common mistakes of the same kind but we as humans can't interpret them. These are the actual situations when we need help of such methods to tell us what's going wrong.
> > > >
> > > > We note that in this line of research, the goal is to explain failure modes in human-understandable terms. So, the focus is to cover and explain failure modes that can be described by humans. We agree with the reviewer that there definitely exists some uninterpretable patterns or situations that the model fails but they are in the scope of our paper.

---

### Author Response · Authors · 2023-11-18

Thanks for the thoughtful reviews. Let us make a few general points that are relevant to several reviews at once.

---
### **G1**: Using state-of-the-art tagging model (RAM) as a part of PRIME

We agree with the reviewers that the tagging model we use is crucial in identifying tags and concepts for failure mode descriptions. However, it is worth noting that any kind of approach that aims to obtain *human-understandable descriptions* for failure modes needs to *utilize an auxiliary model* to bridge the gap between vision and language modalities: e.g. existing works [1, 2, 3] highly depend on the performance CLIP.

We note that we use RAM which is the state-of-the-art tagging model with remarkable performance. In the RAM paper [4], authors observe high-quality precision/recall of their method on different common datasets over different tasks (multi-label classification, segmentation, detection).
These numbers come from Table 2 and Table 3 of [4].

| OPPO common      | OpenImage-common | COCO |
| ----------- | ----------- | ----------- |
| $78.8\\%\\ $/$\\ 79.4\\%$ | $80.3\\%\\ $/$\\ 75.7\\%$ | $82.9\\%\\ $/$\\ 66.4\\%$ |

While the results above show RAM is highly performant generally, we thank the reviewers for their suggestion to verify that the model is similarly effective under the setting we deploy it. To this end, we ran a small-scale human validation study on Living17 (one of the main datasets we used in our paper) to evaluate precision/recall of RAM. We took $100$ images and evaluated tags over these images where we observed $86.85\\%$ for precision and $81.85\\%$ for recall.


Additionally, our use of the auxiliary model (tagging models) is *closely aligned* to the task they are optimized for. That is, these models are trained to efficiently detect various objects and concepts in images, assigning tags accordingly. As tagging models advance, PRIME's effectiveness is expected to be enhanced. Also, we believe it is highly likely that tagging models like RAM that are finetuned on *specific domains* will arise very soon, similar to how ConVIRT (CLIP for medical data) came about soon after CLIP. Presently, one can fine-tune a tagging model for medical image-text datasets using the provided finetuning code for RAM.

---
### **G2:** Confusion about using language models for generating captions:
We apologize for the confusion. We only used language models for a very small step in our evaluation procedure where we used text-to-image generative models (Section 4.2). In those experiments, we select some success/failure modes for Living17 classes (more details in Appendix A.9), and generate captions for generative models. We will add a section to the Appendix covering tags and captions. Here we provide a few of them. In our experiments, we inspected generated images and ensured that they are relevant to captions and images inside the failure modes.

class: bear

gray + water → “a photo of a gray bear in water”;

river → “a photo of a bear in the river”

cub + climbing + tree → “a photo of a bear cub climbing a tree”;

black + cub + branch → “a photo of a black bear cub on a tree branch”;

class: ape

black + branch → “a photo of a black ape on a tree branch”;

sky → “a photo of an ape in the sky”;

gorilla + trunk + sitting → “a photo of a gorilla ape sitting near a tree trunk”;

mother → “a photo of a mother ape”;

For all other evaluation parts, we didn’t use descriptions and only put tags in a bag-of-word manner ( mentioned at the end of 3rd paragraph on page 5). For example, to get the description for class ape with tags black + branch, the description we used for evaluating the quality of description in Section 4.3 is “a photo of ape black branch”.
We argue that utilizing language model is unnecessary and we could use this bag-of-word manner for image generation as well. We will remove the use of ChatGPT in updated draft.

---

[1] Eyuboglu, Sabri, et al. "Domino: Discovering systematic errors with cross-modal embeddings." arXiv preprint arXiv:2203.14960 (2022).

[2] Jain, Saachi, et al. "Distilling model failures as directions in latent space." arXiv preprint arXiv:2206.14754 (2022).

[3] d'Eon, Greg, et al. "The spotlight: A general method for discovering systematic errors in deep learning models." Proceedings of the 2022 ACM Conference on Fairness, Accountability, and Transparency. 2022.

[4] Zhang, Youcai, et al. "Recognize Anything: A Strong Image Tagging Model." arXiv preprint arXiv:2306.03514 (2023).

---

### Author Response · Authors · 2023-11-20
**New draft is uploaded**

We have updated the draft to address the points raised by the reviewers during the rebuttal period. Here are the updates:

+ We added the Limitation Section (A.1) to
    + acknowledge the use of the tagging model and its effect on PRIME.
    + provide a discussion on specific domains.
    + outline the limitations of interpreting tags and the use of language models (as mentioned by reviewer dBy1).
+ In Section A.15, we included more detailed information about PRIME hyperparameters, which is referenced at the end of Section 3.2.
+ We added more detailed explanations for image generation experiments in A.10.
+ Correlation coefficients between train and test drop have been added to Section A.5.
+ A human study (on Living17) has been incorporated to evaluate RAM in Section A.14.
+ We included papers provided by reviewers to the related work section.
+ In Sections 4.2 and 4.3, we clarified confusion G2 and explicitly mentioned the bag-of-word manner for evaluating quality of descriptions.
+ Suggestions made by the reviewers to improve the quality of text is also considered.

We hope this new draft can address the reviewers' concerns and result in better scores for this work.

---

### Meta-Review · Area_Chair_hGHj · 2023-12-08

**Metareview:**

This paper proposes a novel approach for identifying and communicating failure modes of ML models for users. The main contribution is instead of identifying the failure modes first, the proposed method starts with identifying human-interpretable tags. In this way, the identified failure modes can be easily understood by humans. The method is very straightforward. The method is validated using large scale datasets and using quantitative metrics.

Strengths: all reviewers recognize the importance of the problem and the idea is novel.

Weaknesses: there are remaining limitations of this work. For example, the tags may not cover rare cases and these failure modes will be ignored. Also, multiple reviewers mentioned potential limitations in the evaluation. There are multiple components in the whole system, including several applications of foundation models and GPT models. There may be accumulated errors that worth more discussion. Finally, there is no human evaluation, given the major focus on human interpretability.

**Justification For Why Not Higher Score:**

The proposed method still has several limitations, as mentioned in the reviews and meta-review. These limitations may be fine for a poster but prevent the paper from reaching a higher bar.

**Justification For Why Not Lower Score:**

The paper's merit outweighs the limitation.

---

### Decision · Program_Chairs · 2024-01-16

Accept (poster)